

# Probabilistic Hydrological Estimation of LandSlides (PHELS): global ensemble landslide hazard modelling

Anne Felsberg[1], Zdenko Heyvaert[1], Jean Poesen[1,2], Thomas Stanley[3,4], and Gabriëlle J. M. De Lannoy[1]

[1]KU Leuven, Department of Earth and Environmental Sciences, Heverlee, Belgium
[2]Maria-Curie Sklodowska University, Faculty of Earth Sciences and Spatial Management, Lublin, Poland
[3]GESTAR II, University of Maryland Baltimore County, Baltimore, USA
[4]Hydrological Sciences Laboratory, NASA Goddard Space Flight Center, Greenbelt, Maryland

**Correspondence:** Anne Felsberg ( anne.felsberg@gmail.com)

**Abstract.** In this study we present a model for the global Probabilistic Hydrological Estimation of LandSlides (PHELS). PHELS estimates the daily hazard of hydrologically-triggered landslides at a coarse spatial resolution of 36 km, by combining landslide susceptibility (*LSS*) and (percentiles of) hydrological variable(s). The latter include daily rainfall, a 7-day antecedent rainfall index (*ARI7*) or root-zone soil moisture content (*rzmc*) as hydrological predictor variables, or the combination of rainfall
and *rzmc*. The hazard estimates with any of these predictor variables have areas under the Receiver Operation Characteristic curve (AUC) above 0.68. The best performance was found with combined rainfall and *rzmc* predictors (AUC=0.79), which resulted in the least amount of missed alarms (especially during spring) and false alarms. Furthermore, PHELS provides hazard uncertainty estimates by generating ensemble simulations based on repeated sampling of *LSS* and the hydrological predictor variables. The estimated hazard uncertainty follows the behaviour of the input variable uncertainties, is about 13.6 % of the
estimated hazard value on average across the globe and in time, and smallest for very low and very high hazard values.

## 1   Introduction

Landslides are mass movements of soil and rock triggered by anthropogenic or seismic activity and, most frequently, by rainfall (Froude and Petley, 2018; Nowicki Jessee et al., 2018; Stanley et al., 2021). In order to limit human and economic losses due to landslides, the prediction of where and when they are likely to occur is crucial (Crozier, 2013). The spatio-
temporal probability of a landslide is generally referred to as 'landslide hazard', and can be estimated based on a range of static environmental (spatial) and dynamic hydrological (temporal) data sources. The spatial and temporal information can either be merged directly, e.g. via machine learning techniques combining rainfall, soil moisture, snow and slope angle in an ad-hoc fashion (Stanley et al., 2020, 2021), or in a two-step process. The last approach is more common and traceable, and requires that spatial and temporal probabilities are estimated individually before combining them into one prediction system.

The spatial probability, referred to as landslide susceptibility (*LSS*), is estimated based on (static) environmental features (Pourghasemi and Rossi, 2016; Reichenbach et al., 2018). Most *LSS* maps are created at the local to regional scale, where they are also used for mitigation and planning purposes (Guzzetti et al., 2005; Crozier, 2013). Others are specifically developed to be used in a landslide early warning system. The global *LSS* assessment by Stanley and Kirschbaum (2017), for instance, has





been developed to become part of the first version of the Landslide Hazard Assessment for Situational Awareness (LHASA)
model (Kirschbaum and Stanley, 2018).

The temporal probability can either be calculated explicitly by physical models that compute the shear strength and stress
in slopes (Whiteley et al., 2019) or approximated by statistical, empirical approaches (Guzzetti et al., 2008, 2020). The latter
relate one or more dynamic hydrological predictor variables to a chance for a landslide (Guzzetti et al., 2008, 2020).

A simple yet effective binary approach is to use thresholds for various measures of rainfall and soil water content beyond
which landslide occurrence is expected (Segoni et al., 2018a). While univariate thresholds in antecedent rainfall index (*ARI*)
or surface soil moisture exist (Kirschbaum and Stanley, 2018; Zhuo et al., 2019), most thresholds are based on two or more
variables. The most frequently used thresholds are based on rainfall intensity and duration or variations thereof (Caine, 1980;
Guzzetti et al., 2008; Rossi et al., 2017; Rosi et al., 2021), and hydro-meteorological thresholds (Ponziani et al., 2012; Brocca
et al., 2016; Devoli et al., 2018; Mirus et al., 2018; Thomas et al., 2019; Uwihirwe et al., 2020, 2022). Alternatively, it is
possible to retrieve a continuous triggering probability based on rainfall (Calvello and Pecoraro, 2019), soil moisture measures
(Wicki et al., 2020) or a combination of both (Bordoni et al., 2020). The measures of soil moisture range from antecedent
soil moisture (Mirus et al., 2018; Wicki et al., 2020) and increase in soil saturation (Wicki et al., 2020) to soil moisture of the
day (Bordoni et al., 2020). In comparison to purely rainfall-based landslide likelihood predictions, the inclusion of soil water
content has been found to prevent false alarms, independent of the data source (Ponziani et al., 2012; Segoni et al., 2018b;
Stanley et al., 2021).

To estimate hazard using the two-step process, the temporal probability assessment is combined with spatial *LSS* information.
Monsieurs et al. (2019a, b) for example developed combined *ARI-LSS* thresholds. Kirschbaum and Stanley (2018) adapted the
level of nowcasts (based on a global univariate *ARI*-threshold) according to *LSS*. Bordoni et al. (2020) updated *LSS* according
to whether or not the temporal probability was above 0.5. We comprise all of the above-mentioned approaches under the term
'hazard modelling'.

The available landslide hazard modelling approaches rarely consider the quantification of uncertainty. For *LSS*, uncertainty
information is sometimes provided (e.g. Broeckx et al. (2018), Depicker et al. (2020) and Felsberg et al. (2022b)). For the
temporal aspect, uncertainties have been assigned to rainfall and rainfall-*LSS* thresholds (Rossi et al., 2017; Monsieurs et al.,
2019a). Hartke et al. (2020) created a probabilistic adaptation of the first version of LHASA by using rainfall distributions
instead of deterministic values. This approach increased the number of correctly predicted landslides and decreased the number
of false alarms from high nowcasts. For a physically-based hazard estimate, Canli et al. (2018) proposed to use an ensemble of
rainfall values as input, resulting in an ensemble of predicted hazard values.

In this study we i) investigate the ability of different hydrological predictor variables for global landslide hazard estima-
tion and ii) use ensembles for an uncertainty assessment. We develop the Probabilistic Hydrological Estimation of LandSlides
(PHELS) model. PHELS provides a global coarse-scale (36-km) dynamic landslide hazard simulation with a reliable uncer-
tainty estimate at any time and location, by combining ensembles of *LSS* (Felsberg et al., 2022a, b) and daily information on
hydrological predictor variables. For the latter, we test ensembles of rainfall and an *ARI* based on reanalysis precipitation data,
and root-zone soil moisture content $[\mathrm{m}^3/\mathrm{m}^3]$ (*rzmc*) from a land surface model (LSM). The paper is guided by the following





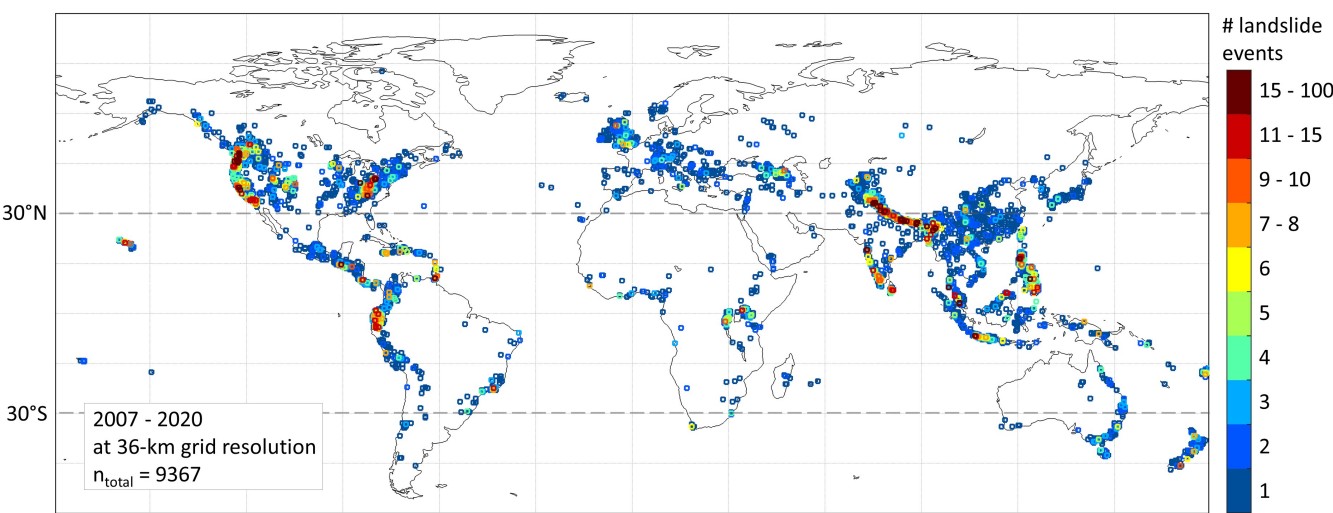

**Figure 1.** Spatial distribution of the number of landslide events per 36-km grid cell for the study period 2007 through 2020. Note the irregular colourbar intervals. Grey dashed lines indicate the latitude stratifications between 'North', 'the tropics' and 'South'.

research questions: 1) Which hydrological variable (or combination of variables) performs best at simulating global landslide

hazards? 2) Is the estimated uncertainty related to the magnitude of the simulated hazard?

## 2   Data, model and methods

### 2.1   Landslides

Despite the existence of many different types of landslides and their manifold shapes and sizes, in this study the term 'land-slide' refers to all types of hydrologically-triggered mass movements. We use landslide data from the most recent version of the

Global Landslide Catalog (GLC) (https://landslides.nasa.gov/viewer, accessed 20th January 2022). This inventory is based on media reports (Kirschbaum et al., 2010, 2015), and supplemented with the citizen science-based Landslide Reporter Catalog (LRC) data (Juang et al., 2019), see Stanley et al. (2021) for details. We select all landslides triggered by 'continuous rain', 'downpour', 'monsoon', 'flooding', 'rain' and 'tropical cyclone' (GLC classifiers). About 99 % of these landslides were collected in the time period 2007 through 2020. We therefore select this time period for the research conducted in this study. Note

that the known economic and English-language bias will affect the completeness of these inventories and reduce the reliability of their 'absence reporting'.

Landslide information from the GLC was already successfully used for a number of global applications. Stanley and Kirschbaum (2017), Lin et al. (2017) and Felsberg et al. (2022b) used the locations of landslides to create global *LSS* maps.



The first version of the global LHASA model was evaluated with landslide data from the GLC (Kirschbaum and Stanley, 2018)
and LHASA version 2.0 was trained with a gridded GLC version (Stanley et al., 2021).

In this study, we additionally included 183 landslides from Russian quarterly reports (FSBIH, 2018; Felsberg et al., 2021)
from 2010 to 2018, but for simplicity, we refer to the combined landslide dataset as 'GLC'. Multiple landslide occurrences on
one day within one grid cell are merged into one landslide event (LSE), resulting in a total number of 9367 LSE for the study
period as displayed in Figure 1.

## 2.2 Landslide susceptibility (*LSS*)

*LSS* describes the spatial likelihood of landslide occurrence (Crozier, 2013). In this study, we use the global *LSS* map of
Felsberg et al. (2022a, b), developed on the 36-km Equal-Area Scalable Earth version 2 (EASEv2) grid for the purpose of sub-
sequent combination with coarse scale (satellite- or model-based) soil moisture data and with extended uncertainty assessment.
This data-driven *LSS* focuses on hydrologically-triggered landslides, and the most prominent predictor variables are the com-
pound topographic index, long-term median surface soil moisture and evaporation, slope-related variables and the peak ground
acceleration. The *LSS* estimates consist of an ensemble of 2500 values per grid cell to reflect the *LSS* probability distribution.
To facilitate subsequent flexible sampling from these ensembles, we fit a beta distribution $\mathcal{B}_g(\alpha_g, \beta_g)$ with shape parameters
$\alpha_g, \beta_g$ to the 2500 ensemble *LSS* realizations at each grid cell $g$, using the package *fitdistrplus* of R version 4.0.3 (R Core Team,
2020) to estimate optimal parameters via maximum likelihood estimation.

## 2.3 Hydrological variables

For the hydrological predictor variables, we derive daily 36-km rainfall data (comprising convective and large-scale liquid
precipitation) and the associated 7-day antecedent rainfall index [mm] (*ARI7*) from the global reanalysis data product Modern-
Era Retrospective analysis for Research and Applications, Version 2 (MERRA-2) (Gelaro et al., 2017), available from 1980
onward. The *ARI7* for day $t$ was introduced by Kirschbaum and Stanley (2018) as a weighted ($w_t$) average of antecedent
rainfall ($r_t$) during the preceding 7 days:

$$ARI7 = \frac{\sum_{t=0}^{6} r_t \cdot w_t}{\sum_{t=0}^{6} w_t} \quad \text{where} \quad w_t = (t+1)^{-2} \tag{1}$$

The MERRA-2 data have a native spatial resolution of 0.625°lon × 0.5°lat and are interpolated to the 36-km EASEv2 grid
via bilinear interpolation. These interpolated MERRA-2 data are also used as input to the state-of-the-art, physically-based
Catchment Land Surface Model (CLSM) (Koster et al., 2000) to simulate *rzmc* (0-100 cm) for the study period. The *rzmc* is
informative of water content at typical depths of most (not all) landslide shear planes and contains information on both surface
water content and groundwater.

CLSM simulations are run with 24 ensemble members by perturbing meteorological input (including rainfall) and select
state variables (see Felsberg et al. (2021)). The resulting ensemble average of rainfall and *rzmc* is used for deterministic hazard
modelling, whereas the ensemble average and standard deviation are used to sample input values for the ensemble hazard
modelling.



In a next step, the sampled hydrological variables are transformed into percentiles to detach their magnitudes from the local climatological conditions. Felsberg et al. (2021) moreover found that the transformation into percentiles of soil water content enhanced the ability to distinguish between LSE and sampled days with no landslide event (noLSE). The climatological percentile thresholds are computed per grid cell based on long-term (entire study period 2007-2020) time series of ensemble
mean simulations of soil water content, similar as in Felsberg et al. (2021).

## 2.4 The PHELS model

The objective of PHELS is to obtain a measure of landslide hazard in a probabilistic way. The probability of landslide event occurrence ($LSE = 1$) given static environmental conditions or dynamic variables $x_i$ can be described stochastically through conditional probabilities $p(LSE = 1 \mid x_i)$ (van Westen et al., 2006; Calvello and Pecoraro, 2019; Uwihirwe et al., 2020; Lom-
bardo et al., 2020; Felsberg et al., 2021). For the static condition we use *LSS* ($x_L$), and for the dynamic variable we use percentiles of daily *rzmc*, rainfall and *ARI7* describing the hydrological conditions ($x_h$). The probability of a landslide event occurring conditioned on the susceptibility of the location and the hydrological state of the day can be defined as follows using Bayes' law:

$$p(LSE = 1 \mid x_L, x_h) = \frac{p(x_L, x_h \mid LSE = 1) \cdot p(LSE = 1)}{p(x_L, x_h)} \tag{2}$$

or if two hydrological variables are taken into account:

$$p(LSE = 1 \mid x_L, x_{h1}, x_{h2}) = \frac{p(x_L, x_{h1}, x_{h2} \mid LSE = 1) \cdot p(LSE = 1)}{p(x_L, x_{h1}, x_{h2})} \tag{3}$$

Bayes' theorem connects the *prior* probability $p(LSE = 1)$ with a known *likelihood function* of the conditions $p(x_L, x_h \mid LSE = 1)$ to obtain a *posterior* probability. While *LSS* could conceptually be considered a prior probability we opted to use it as a temporally static (but spatially varying) variable and implement it in a similar way as the temporally dynamic soil
moisture. In this study, $p(LSE = 1)$ thus remains an uninformative prior, that is assumed constant in space and time. Since we use percentiles of *rzmc*, rainfall and *ARI7* as hydrological predictor variables, their respective distributions are (quasi-)uniform. For simplicity, we omit the normalizing joint probability terms and use the following proportionality approach:

$$p(LSE = 1 \mid x_L, x_h) \propto p(x_L, x_h \mid LSE = 1)$$
$$\Rightarrow \mathcal{H} = f(x_L, x_h) \tag{4}$$

or for multiple hydrological variables $\mathcal{H} = f(x_L, x_{h1}, x_{h2})$, i.e. a function of the predictor variables $x_L$, $x_{h1}$ and $x_{h2}$. We
refer to this posterior probability value as landslide hazard [-] ($\mathcal{H}$). By foregoing the normalization, and because the absolute values of the distribution fits (below) depend on the binning and dimensions (scale) of the underlying data, the absolute values of $\mathcal{H}$ with one or two hydrological variables will not be comparable. However, for hazard estimation, only a relative spatio-temporal assessment is of importance.

To estimate $p(x_L, x_h \mid LSE = 1)$ we extract values of *LSS* and the hydrological variables for the 9367 LSE. Figure 2a
shows the bivariate histogram for percentiles of *rzmc*, with the number of LSE indicated in color. Distributions for the other hydrological variables (not shown) generally exhibit the same behaviour: LSE are exponentially more likely to occur where *LSS* is high and under wet hydrological conditions, both individually and combined. At the same time, low *LSS* or drier





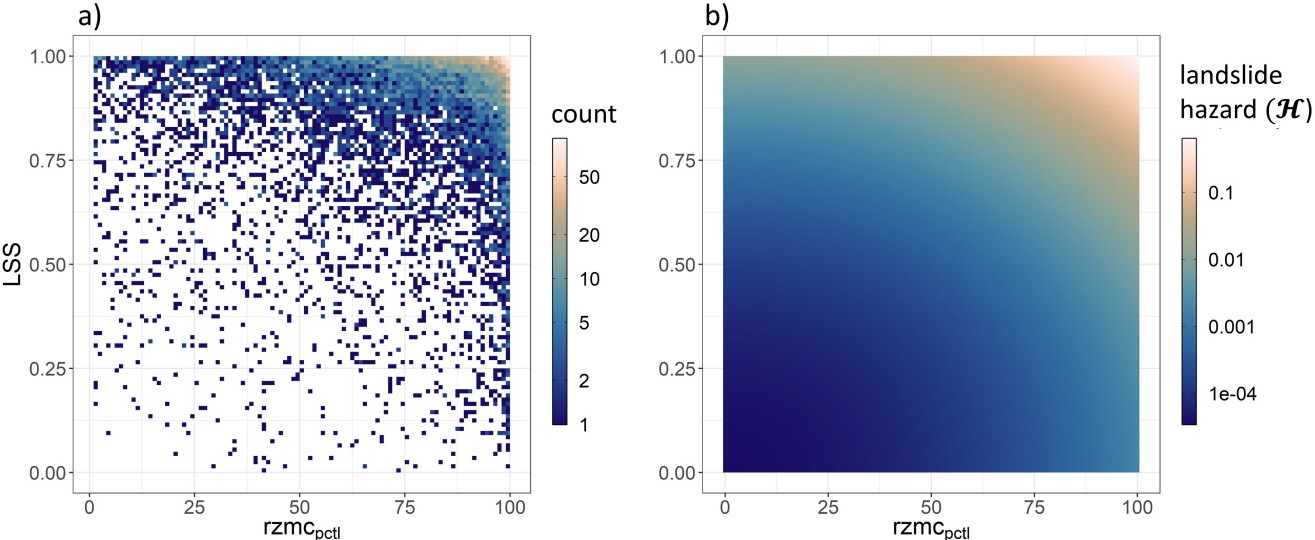

**Figure 2.** a) Bivariate histogram of daily percentiles of *rzmc* ($rzmc_{pctl}$) and *LSS* for landslide events between 2007 and 2020. This data was used to fit the bivariate exponential of equation 5. b) Hazard ($\mathcal{H}$ [-]) as a bivariate exponential fitted function of daily $rzmc_{pctl}$ and *LSS*. Note the logarithmic colourbar intervals.

conditions do not exclude the possibility of landslide occurrence. We find that 1.8%/1.3%/0.7% of the LSE occurs where *LSS* is below 0.5 [-] and percentiles of *rzmc*/rainfall/*ARI7* are below 50 [-]. For very susceptible locations, LSE also occur at much

drier hydrological conditions, whereas very wet conditions still require a certain level of *LSS*. This results in a skewing of the distribution from the upper right towards the upper left corner, which was also observed by Monsieurs et al. (2019a).

Next, a two- or three-dimensional quadratic-exponential function is fit through the extracted LSE data. This kind of fit through the distribution of data points is a long-term and spatially aggregated summary statistic, also referred to as system signature (Vrugt and Sadegh, 2013). We tested different forms of the fitting equation and found the lowest root mean squared

deviation (in reference to the full LSE distribution) for the following exponential functions with one and two hydrological predictor variables, respectively:

$$\mathcal{H} = a \cdot \exp(b \cdot x_L^2 + c \cdot x_h^2) \tag{5}$$

and

$$\mathcal{H} = a \cdot \exp(b \cdot x_L^2 + c \cdot x_{h1}^2 + d \cdot x_{h2}^2) \tag{6}$$

These equations are the core of the PHELS model. Ensuring $\sum_{x_L, x_h} \mathcal{H} \equiv 1$ for $x_L \in \{0, 0.01, .., 1\}$ (binned continuous values) and $x_h \in \{1, 2, .., 100\}$ (percentiles, discrete), results in the parameters shown in Table 1. Note that ensuring the sum of 1 only offsets the scaling factor $a$ and does not affect the other parameters. For the fitting ('nls') and summing, we use R



**Table 1.** Parameters of the exponential fit (Equations 5 and 6) for PHELS based on different hydrological variables (columnwise). Parameters are given for the static $x_L$ (parameter b) and one hydrological variable $x_h$ (parameter c for *rzmc*, rainfall or *ARI7*) or two hydrological variables $x_{h1}, x_{h2}$ namely *rzmc* (parameter c) and rainfall (parameter d). To simplify comparison between parameters b, c and d across the different orders of magnitudes ($\mathcal{O}(x_L^2) = 1$ and $\mathcal{O}(x_h^2) = 10000$), c and d are shown as multiples of $10^{-4}$. Residual standard errors are shown for all fits, as well as the theoretical maximum hazard values $\mathcal{H}_{max} = \mathcal{H}(x_L = 1, x_h = 100)$ and $\mathcal{H}_{max} = \mathcal{H}(x_L = 1, x_{h1} = 100, x_{h2} = 100)$, respectively. Note that the absolute $\mathcal{H}$ are not to be compared for PHELS models with varying numbers of input variables.

| Parameters | rainfall | *ARI7* | *rzmc* | *rzmc*&rainfall |
|---|---|---|---|---|
| a [$\times 10^{-5}$] | 5.33 | 3.32 | 3.45 | 6.10 |
| b [1] | 5.18 | 5.19 | 5.91 | 0.48 |
| c [$\times 10^{-4}$] | 4.16 | 4.78 | 3.96 | 0.50 |
| d [$\times 10^{-4}$] | - | - | - | 0.41 |
| Residual standard error | 0.037 | 0.037 | 0.038 | $9.98 \times 10^{-5}$ |
| $\mathcal{H}_{max}$ | 0.67 | 0.60 | 0.71 | $2.43 \times 10^{-4}$ |

version 4.0.3 (R Core Team, 2020), and optimal parameters are obtained by minimizing the residual sum-of-squares between observed and fitted counts.

The difference between parameters for $x_L^2$ and $x_h^2$ reflects the observed skew in the bivariate histogram. The skew in $\mathcal{H}$ is most pronounced as a function of *rzmc*, reduces for rainfall and is least for *ARI7*. When having both *rzmc* and rainfall as hydrological predictor variables (referred to as *rzmc*&rainfall), *rzmc* and *LSS* become equally important and rainfall slightly less. This indicates that both *LSS* and soil wetness are necessary preconditions for LSE occurrence. The different order of magnitude in the parameters of PHELS based on Equations 5 and 6 is a result of extending the quadratic exponential to a third

predictor variable. Retaining the integral of 1 over the predictor space moreover reduces the magnitude of resulting $\mathcal{H}$ from maximum values of $\sim 0.7$ to 0.00024 (see Table 1). This effect can be avoided by using the complete Bayesian theorem as in Equations 2-3 with inclusion of a normalization of the probability instead of the proportionality approach of Equation 4. The average residual standard error is $\sim 0.04$ when using only *rzmc*, rainfall or *ARI7* as predictor along with *LSS*, and the error is relatively larger when two hydrological variables are included. This is because a multidimensional fit is harder to achieve (more

variation to account for). Nevertheless, the resulting distribution of $\mathcal{H}$ as shown in Figure 2b for *rzmc* percentiles represents the observed patterns (Figure 2a) well.

    PHELS estimates can be obtained for single values of the hydrological predictor variables and *LSS*. Such PHELS estimates are referred to as deterministic $\mathcal{H}$. In order to propagate uncertainties of these input variables, PHELS can also be run as an ensemble with members $i = 1, ..., N_{ens}$. Figure 3 illustrates this approach for one grid cell ($g$) at one timestep ($t$ [days]) for $\mathcal{H}$

based on *rzmc*, rainfall and *LSS*. First, we sample *LSS* from the beta distribution $\mathcal{B}_g(\alpha_g, \beta_g)$, and obtain $x_{L,g,i}$. Next, we sample $rzmc_{g,t,i}$ from a normal distribution where $\mu_{g,t}$ and $\sigma_{g,t}$ are the *rzmc* ensemble average and standard deviation diagnosed





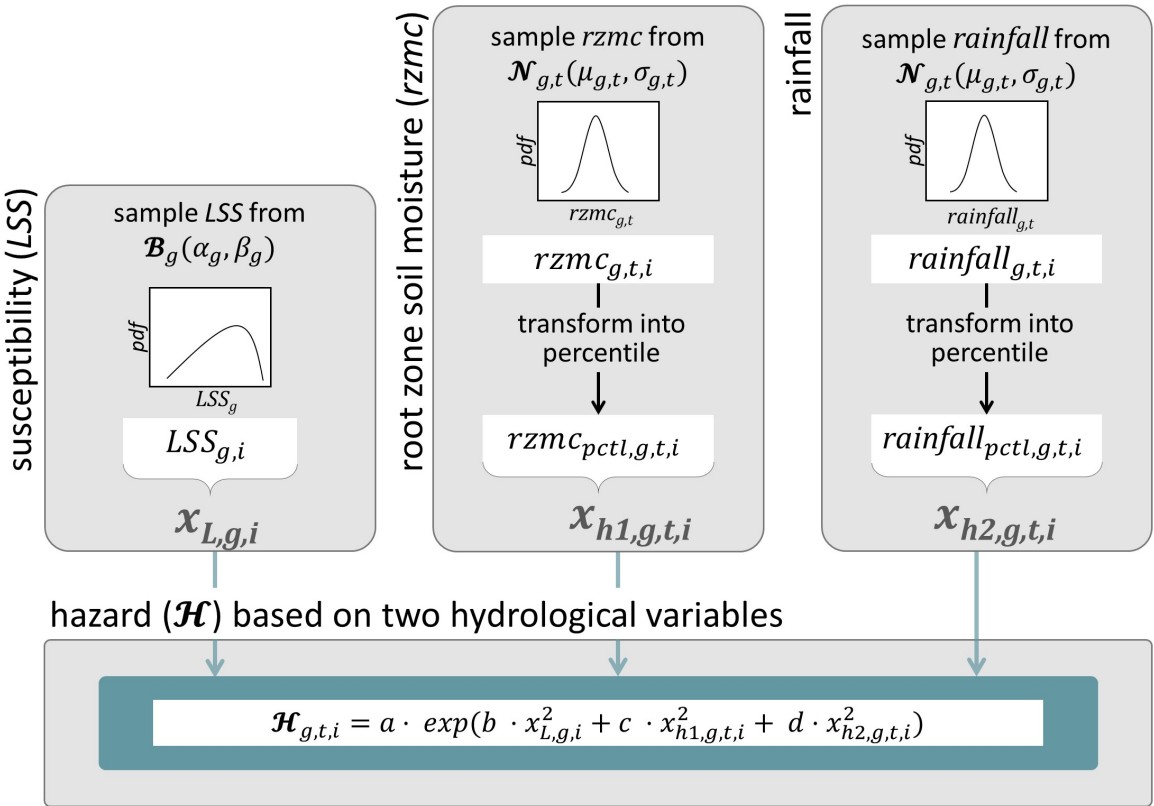

**Figure 3.** Schematic of the sampling setup within PHELS for the example of combined hydrological predictor variables *rzmc*&rainfall. $i$ refers to the ensemble member, $g$ to the grid cell and $t$ to the time [days]. Percentiles of *rzmc* and rainfall on day $t$ are used to derive hazard estimates $\mathcal{H}$ for the same day following Equation 6. $\mu$ and $\sigma$ are the ensemble average and standard deviation. The transformation into percentiles is achieved by comparing $rzmc_{g,t,i}$ and rainfall$_{g,t,i}$ against long term percentile thresholds of the corresponding grid cell $g$.

from 24 ensemble CLSM simulations. The sampled $rzmc_{g,t,i}$ is subsequently transformed into the corresponding percentile by comparison against long-term percentile thresholds for this grid cell and used as $x_{h,g,t,i}$ from Equation 5, or as $x_{h1,g,t,i}$ from Equation 6. The same way of sampling is used for rainfall or *ARI7*. Applying Equation 6 to $x_{L,g,i=1}$, $x_{h1,g,t,i=1}$ and

$x_{h2,g,t,i=1}$ yields the first hazard ensemble member $\mathcal{H}_{g,t,i=1}$. The sampling is repeated $N_{ens} = 100$ times to retrieve a landslide hazard ensemble ($\mathcal{H}_{ens}$). This allows us to obtain an ensemble average $\mathcal{H}$ ($\overline{\mathcal{H}}$) with a connected uncertainty (ensemble standard deviation). Note that all ensemble sampling was performed independently in time, and for each variable, without accounting for temporal autocorrelations or crosscorrelations between variables. During hydrological extreme events such as tropical storms, this may result in conservatively high $\mathcal{H}$ uncertainty estimates in comparison to sampling from multidimensional distributions

(not shown). PHELS is coded in R version 4.0.3 (R Core Team, 2020). A 1-day simulation of the global $\mathcal{H}_{ens}$ with $N_{ens} = 100$ at 36-km spatial resolution takes $\sim$5 minutes on one core.



## 2.5 Evaluation

The evaluation of PHELS is performed both for deterministic and ensemble $\mathcal{H}$ estimates. The strength of PHELS is that it provides relative estimates of hazards in both space and time. However, given the known strong spatial performance of the *LSS* (Felsberg et al., 2022b), the focus will be on a conservative evaluation of the performance in time.

More specifically, the PHELS hazard results are evaluated at grid cells and days of LSE for the study period (2007-2020, $n_{total} = 9367$). For grid cells with at least one LSE, we randomly sample an equal amount of values for noLSE from all other timesteps. In order to account for possible errors in the date reporting and time-zone matching of observations and hydrological data (local time versus UTC), 3 days prior and after an LSE are excluded from the selection as noLSE. For the same reasons we also evaluate the performance for maximum hazard values within a 3 day window around the LSE ($\pm 1$ day, 'LSE3') as was done by Kirschbaum and Stanley (2018) and Monsieurs et al. (2019a). To evaluate the full spatio-temporal performance, we moreover test the performance when $n_{total}$ noLSE are randomly sampled across the globe and in time, without restriction to the LSE grid cells ('noLSEglobal').

We evaluate the performance of the PHELS models with various hydrological predictor variables (*rzmc*, rainfall, *ARI7* and *rzmc*&rainfall). The resulting $\mathcal{H}$ is compared for LSE and noLSE in terms of Receiver Operation Characteristic (ROC) curve, where the true positive rate (TPR) is plotted against the false positive rate (FPR) for different thresholds in the continuous probability values of $\mathcal{H}$. The TPR is the ratio of correctly predicted LSE ('true positives') to the total number of LSE (Wilks, 2011). An LSE is assumed to be predicted when the probability is above a set threshold. The FPR is the ratio of erroneously predicted LSE ('false positives') to the total number of noLSE, here being the same as LSE due to our 1:1 ratio of sampling. For a perfect prediction, the area under the ROC curve (AUC) is 1. A value of 0.5 on the other hand indicates that the prediction is not better than a uniform random prediction. We conduct the ROC analysis for the full data sets of LSE and noLSE, and for LSE3 and noLSEglobal, as well as subsets stratified for latitude and season. Latitudes are stratified at 30°N and 30°S into 'North', the 'tropics' and 'South', as indicated by the dashed lines in Figure 1. Note that the 'South' subset contains much less data. Stratification for seasons follows meteorological standards, i.e. December-January-February (DJF), March-April-May (MAM), June-July-August (JJA) and September-October-November (SON). For additional insight, we compare numbers of false predictions, i.e. false alarms ('false positives') and missed alarms ('false negatives') for the LSE grid cells only. We set the $90^{th}$ percentile of $\mathcal{H}$ within one grid cell over all timesteps as a threshold to distinguish between predicted positive and negative.

## 3 Results

### 3.1 Deterministic hazard estimates

Figure 4 shows time series of PHELS $\mathcal{H}$ in a grid cell near Seattle, USA, based on different ensemble mean predictor variables as deterministic input. For rainfall, fast changes at rainfall events induce a spiky pattern with frequent short-term changes in $\mathcal{H}$. For *ARI7* fewer spikes are visible and the rainy season stands out more. Seasonal patterns are even more pronounced for *rzmc*





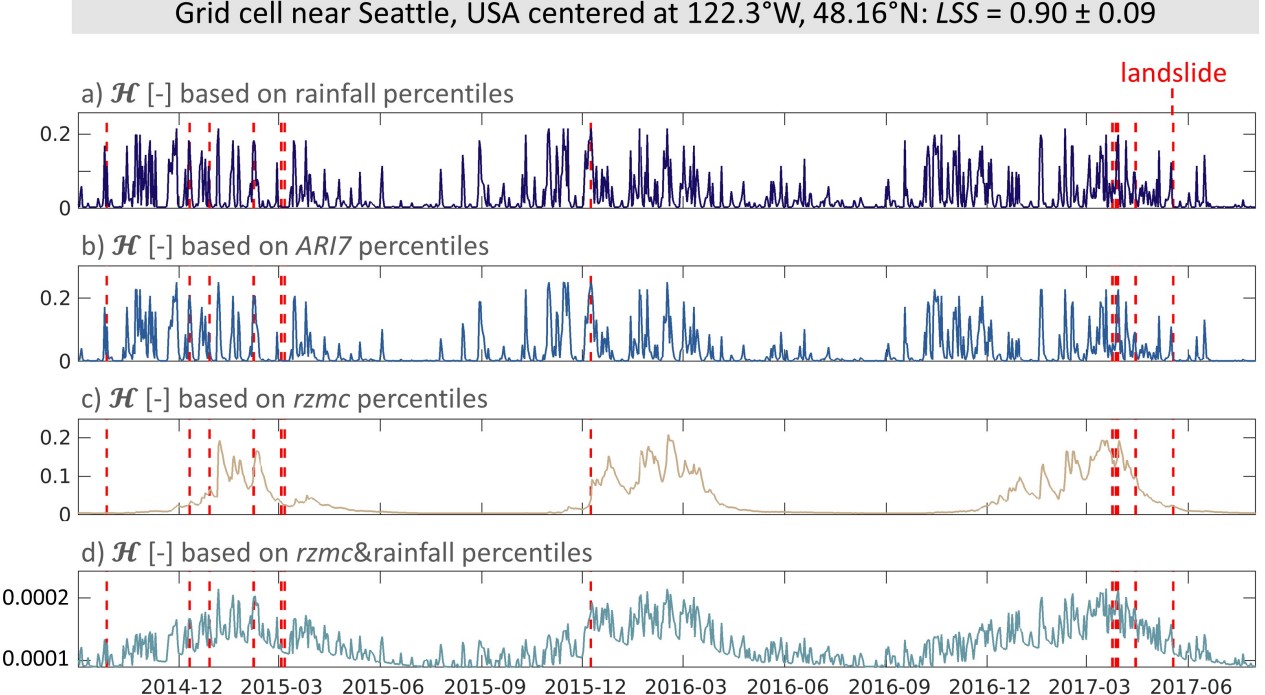

**Figure 4.** Time series of daily PHELS hazard ($\mathcal{H}$ [-]) based on different hydrological predictor variables for a grid cell near Seattle, USA. $\mathcal{H}$ is based on percentiles of a) rainfall, b) *ARI7*, c) *rzmc*, d) *rzmc*&rainfall. Note the different scale in magnitude for d). Days with LSE are indicated by the red dashed lines.

reflecting longer-term transitions, while the signal of the rainfall event spikes is strongly dampened. $\mathcal{H}$ based on *rzmc*&rainfall

shows both the general long-term seasonality and short-term spikes of rainfall events. In this grid cell, LSEs usually coincide with peaks or higher values in simulated $\mathcal{H}$. The first LSE in autumn 2014 coincides with a peak in $\mathcal{H}$ based on rainfall and *ARI7*. Based on *rzmc*, $\mathcal{H}$ is however very small and would have resulted in a missed alarm. The opposite is the case for two LSEs in March 2015 where $\mathcal{H}$ based on *rzmc* is still elevated, but close to zero for rainfall and *ARI7*. When based on *rzmc*&rainfall, both examples show elevated $\mathcal{H}$.

Figure 5 shows ROC curves for $\mathcal{H}$ estimates from PHELS with different hydrological predictor variables, yielding AUC values between 0.68 and 0.79, only for grid cells with at least one LSE in their time history, and only considering the $\mathcal{H}$ estimates at the exact days of the LSE and noLSE samples. $\mathcal{H}$ based on *rzmc*&rainfall performs better than based on one single hydrological variable, closely followed by $\mathcal{H}$ based on *ARI7* (difference of only 0.02).



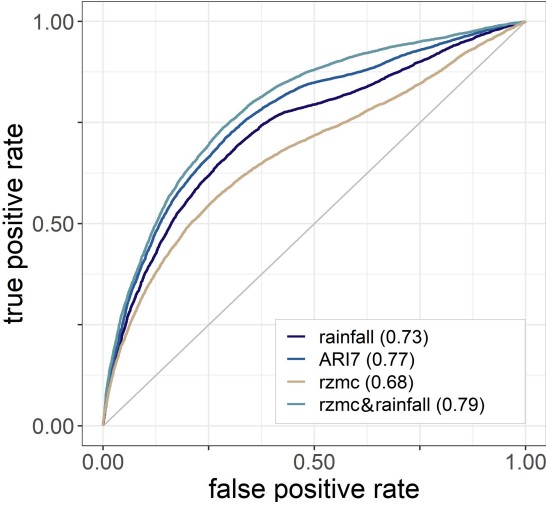

**Figure 5.** ROC curves for PHELS $\mathcal{H}$ simulations using one hydrological variable (rainfall, *ARI7*, *rzmc*) or both *rzmc*&rainfall based on the original LSE and noLSE samples, i.e. for the reported dates and only within LSE grid cells. AUC values are indicated in brackets.

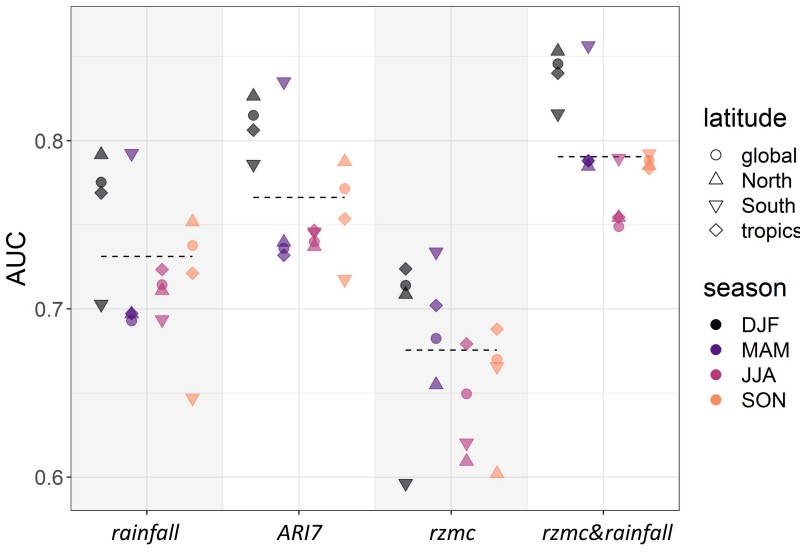

**Figure 6.** AUC values for PHELS hazard stratified per latitude and season, for the different hydrological predictor variables. AUC of the full data is indicated by the horizontal dashed line. Season stratifications are shown from left to right (and dark to light color) per predictor variable: DJF, MAM, JJA, SON. This AUC analysis is based on the LSE and noLSE samples for the reported dates and only within LSE grid cells.



To disentangle these differences in performance, Figure 6 shows AUC values for different subsets of the LSE-noLSE vali-
dation data per latitude and season. The best performance is typically found for the subset MAM-South across all hydrological
variables. The second best performance is found during DJF-North for rainfall, *ARI7* and *rzmc*&rainfall and during DJF-tropics
for *rzmc*. In general, $\mathcal{H}$ based on *rzmc* performs above average for the tropics throughout all seasons. Apart from JJA-tropics
based on *rzmc*, performances for JJA are below average across all predictor variables and latitude stratifications. During spring
time (MAM-North, SON-South), rainfall and *ARI7* performances are largely below average, *rzmc* performance is closer to the
average and for *rzmc*&rainfall performance is average.

Table 2a-b gives an overview of the hits, misses and false alarms for the LSE-noLSE validation data when setting a threshold
at the $90^{th}$ temporal percentile of deterministic $\mathcal{H}$ in the associated grid cells and individually per hydrological predictor
variable. Considering all LSE-noLSE reference data, the number of hits (misses) is highest (lowest) when $\mathcal{H}$ is simulated using
*ARI7* or *rzmc*&rainfall. The number of false alarms is lowest for $\mathcal{H}$ based on *rzmc*, or second lowest when using *rzmc*&rainfall.
For the subset of MAM-North, which showed the largest difference in performance between the predictor variables (see above),
the number of hits (misses) is highest (lowest) for *rzmc*&rainfall. While this comes at the cost of an increased number of false
alarms, this increase in FPR is outweighed by the increase in TPR.

When using the maximum hazard in a 3-day window around the reported LSE (LSE3) and noLSE, performance in terms
of AUC strongly increases for rainfall (and *ARI7*) so that they become similarly well-performing (see Table 2a-b). For *rzmc*
performance is less impacted, and hazard simulations based on *rzmc*&rainfall are moderately improved. The order of best to
worst performing predictor variable(s) remains the same. In contrast, the order is changed when noLSE are sampled globally
without restriction to LSE grid cells (noLSEglobal). For LSE-noLSEglobal, simulations based on *ARI7* perform best, followed
by those based on rainfall, *rzmc*&rainfall and finally *rzmc*. For all four, the AUC values are however really high ($> 0.93$).

To summarize, PHELS $\mathcal{H}$ estimates are best in the (wet) winter in the North and based on LSE-noLSE PHELS performs
best when using both *rzmc*&rainfall as predictor variables. For the analysis of subsequent ensemble results, we therefore focus
on this model, unless noted otherwise.

## 3.2   Ensemble hazard estimates

Figure 7a shows time series of $\mathcal{H}_{ens}$ obtained by PHELS using ensembles of *rzmc*, rainfall and *LSS*. The 100 ensemble
members are shown in light grey, with $\overline{\mathcal{H}}$ on top. The range of $\mathcal{H}$ between ensemble members is largest for high $\overline{\mathcal{H}}$. This
translates into large ensemble standard deviations for high $\overline{\mathcal{H}}$, shown in Figure 7b. The ensemble standard deviation serves as
a measure of simulation uncertainty. We observe a peak in the uncertainty for the three first LSE that occur at peaks in $\overline{\mathcal{H}}$. For
the two LSE observed in this grid cell in spring 2015, there is no peak in $\overline{\mathcal{H}}$ nor in the uncertainty.

Table 2c gives the AUC values for $\overline{\mathcal{H}}$, as well as the hits, misses and false alarms. Compared to deterministic PHELS for the
same hydrological predictor variables, we observe a very minor increase in performance.

To understand the spatio-temporal behaviour of PHELS simulations, Figure 8 shows spatial distributions of PHELS input
data (ensemble average *rzmc*, rainfall and *LSS*) and the resulting $\overline{\mathcal{H}}$ for 15 September 2015. We find elevated $\overline{\mathcal{H}}$ in Central
America, western Subsaharan Africa, central India and China. These hotspots are located where high values of all three input



**Table 2.** Performance of deterministic PHELS $\mathcal{H}$ based on different hydrological predictor variables $x_h$ for a) all LSE and noLSE validation data and b) the subset during MAM north of 30°N. Shown are the AUC, numbers of hits, misses and false alarms, as well as the TPR and FPR. Additional AUC values were calculated when accounting for possible temporal offset in the dating of the LSE within a 3 day window (±1 day, LSE3) as well as for noLSE sampling without constraint to grid cells of LSE occurrence (noLSEglobal). For comparison c) shows performance of PHELS ensemble average $\overline{\mathcal{H}}$ for original LSE and noLSE validation data.

a) *deterministic $\mathcal{H}$, all data*

| $x_h$ | hits | misses | false alarms | TPR | FPR | AUC | AUC LSE3 | AUC noLSEglobal |
|---|---|---|---|---|---|---|---|---|
| **rainfall** | 4201 | 5163 | 943 | 0.449 | 0.101 | 0.731 | 0.804 | 0.944 |
| **ARI7** | 4597 | 4767 | 941 | 0.491 | 0.100 | 0.766 | 0.813 | 0.950 |
| **rzmc** | 3656 | 5708 | 905 | 0.390 | 0.097 | 0.675 | 0.691 | 0.930 |
| **rzmc&rainfall** | 4338 | 5026 | 931 | 0.463 | 0.099 | 0.791 | 0.836 | 0.936 |

b) *deterministic $\mathcal{H}$, MAM-North*

| $x_h$ | hits | misses | false alarms | TPR | FPR | AUC | AUC LSE3 | AUC noLSEglobal |
|---|---|---|---|---|---|---|---|---|
| **rainfall** | 507 | 688 | 104 | 0.424 | 0.100 | 0.697 | 0.778 | 0.939 |
| **ARI7** | 573 | 622 | 93 | 0.479 | 0.089 | 0.740 | 0.789 | 0.948 |
| **rzmc** | 669 | 526 | 167 | 0.560 | 0.161 | 0.655 | 0.666 | 0.941 |
| **rzmc&rainfall** | 736 | 459 | 173 | 0.616 | 0.166 | 0.785 | 0.837 | 0.943 |

c) *ensemble average $\overline{\mathcal{H}}$, all data*

| $x_h$ | hits | misses | false alarms | TPR | FPR | AUC |
|---|---|---|---|---|---|---|
| **rzmc&rainfall** | 4386 | 4978 | 940 | 0.468 | 0.100 | 0.792 |





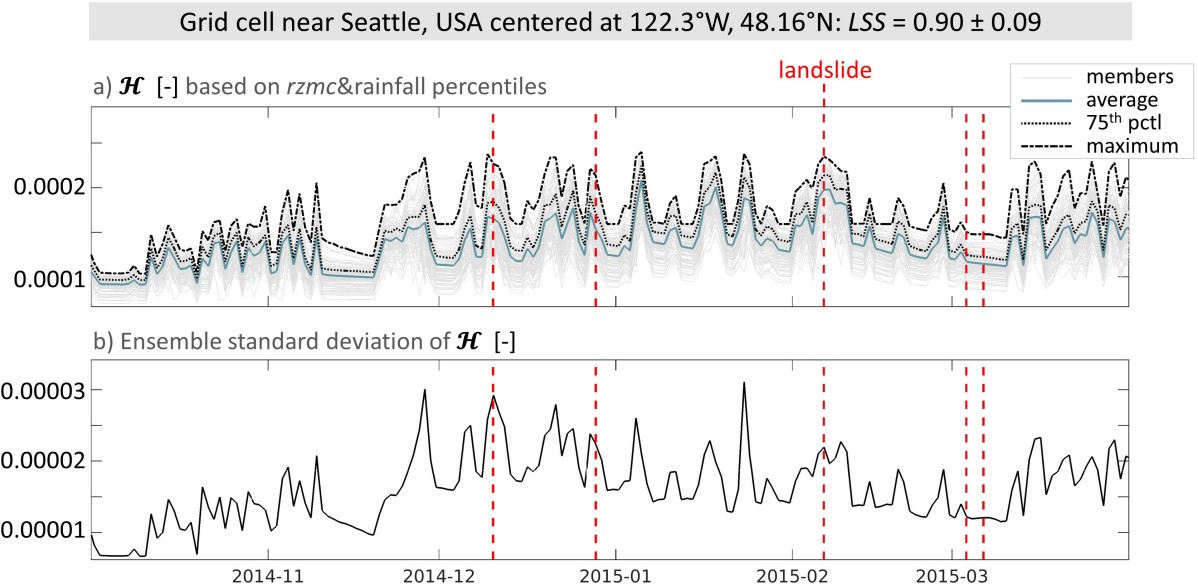

**Figure 7.** Time series of daily PHELS hazard ensembles ($\mathcal{H}_{ens}$) based on *rzmc*&rainfall for a grid cell near Seattle, USA (same as in Figure 4). (a) ensemble members as well as ensemble average $\overline{\mathcal{H}}$, $75^{th}$ percentile and maximum, b) ensemble standard deviation. Days with LSE are indicated by the red dashed lines.

variables coincide. Note that $\mathcal{H}$ uses percentiles of *rzmc* and rainfall, but that Figure 8a-b show absolute *rzmc* and rainfall values, not percentiles. Large *rzmc* values do therefore not necessarily indicate extremely wet conditions for a specific location.

Example of this are the northern hemisphere peat areas in central Siberia or close to the Hudson Bay, which are generally wetter than other regions, but are not necessarily wetter than normal on the shown date.

The ensemble standard deviations of PHELS input and output are shown in Figure 9 for the same day. As for the ensemble averages, we find a high (low) ensemble standard deviation of $\mathcal{H}$, i.e. uncertainty, where high (low) uncertainty of the input variables coincide. Example areas with high uncertainty are Central America and China. Example areas with low uncertainty are

are the central USA, the Amazon and the Congo basin.





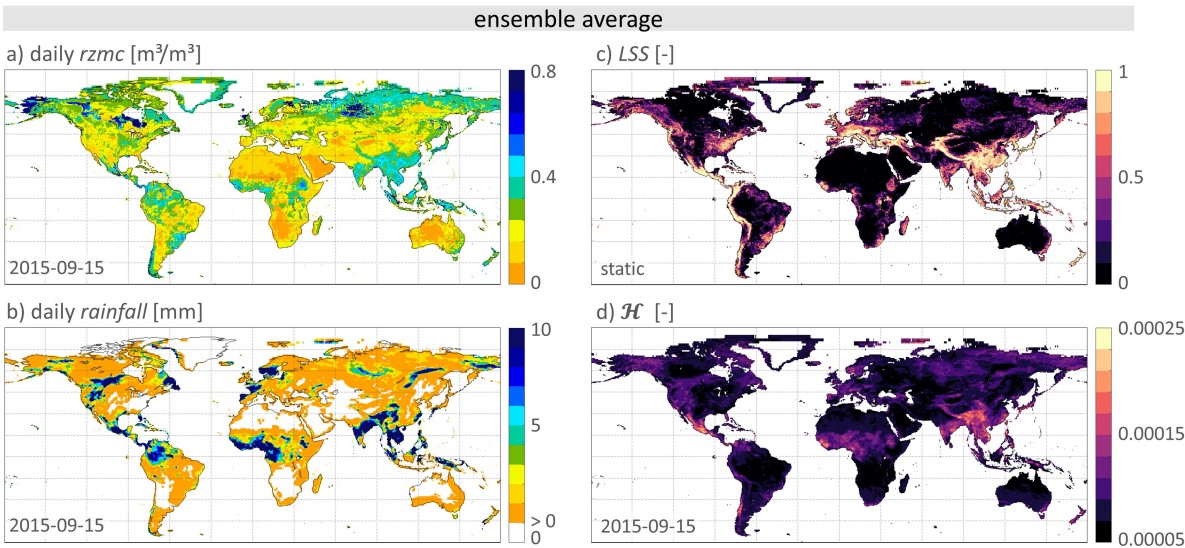

**Figure 8.** Spatial distribution of PHELS input and output for 15 September 2015. Shown are the grid cell's ensemble average a) *rzmc* [m³/m³], b) rainfall [mm], c) static *LSS* [-] and d) hazard $\mathcal{H}$ [-].

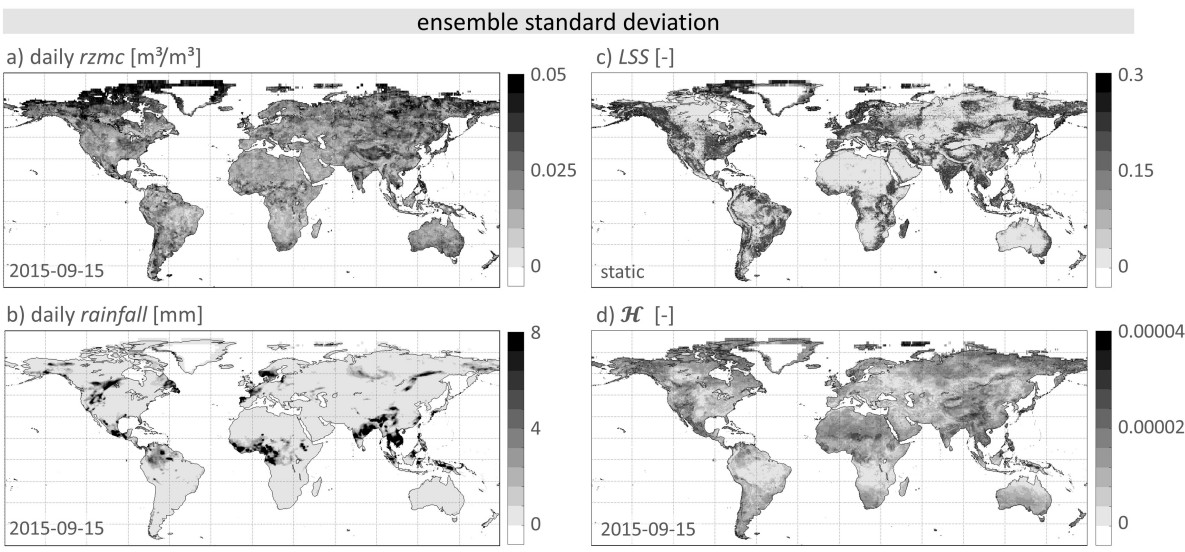

**Figure 9.** Spatial distribution of PHELS input and output for 15 September 2015. Shown are the grid cell's ensemble standard deviations of a) *rzmc* [m³/m³], b) rainfall [mm], c) static LSS [-] and d) hazard $\mathcal{H}$ [-].





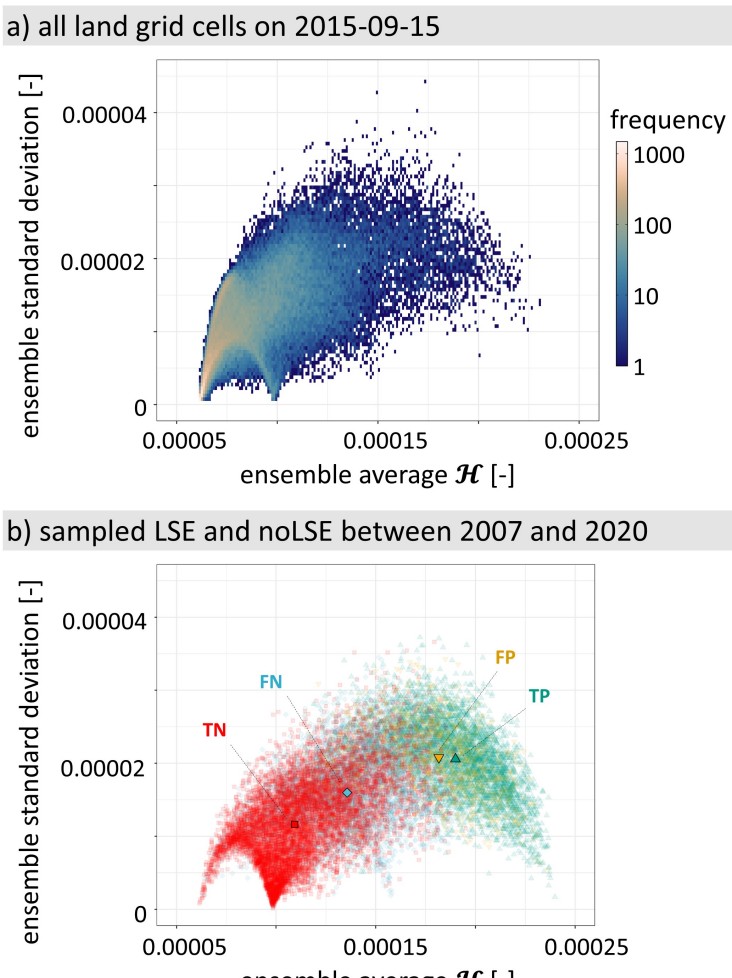

**Figure 10.** Bivariate histogram of ensemble average $\overline{\mathcal{H}}$ against ensemble standard deviation a) for all grid cells on 15 September 2015 and b) for LSE and noLSE during the complete study time period. Colors and shapes indicate whether the predictions are correct (true negative: TN, true positive: TP) or not (false negative: FN, false positive, FP). Median values are indicated by the larger symbols on top of the scatter cloud. As a threshold for $\overline{\mathcal{H}}$ predictions to count as LSE or noLSE, we use the $90^{th}$ percentile of $\mathcal{H}$ per grid cell.

The relationship between $\overline{\mathcal{H}}$ and its uncertainty in terms of standard deviation is illustrated in Figure 10a for all land grid cells on 15 September 2015. A parabolic pattern between low and high $\overline{\mathcal{H}}$ values is found, where uncertainty is low at either end and increased in the middle. Furthermore, a more linear trend ('tail') is found where uncertainty increases proportionally to $\overline{\mathcal{H}}$. Figure 10b shows $\overline{\mathcal{H}}$ and standard deviation of $\mathcal{H}_{ens}$ for all LSE and noLSE. Symbols and color indicate whether LSE

and noLSE were correctly captured (true positive - TP; true negative - TN) or not (false positive - FP; false negative - FN). The median uncertainty for FN, i.e. missed alarms, is larger than that for TN. Median uncertainty for FP, i.e. false alarm, is however



nearly identical to that of TP. High uncertainty is therefore not necessarily an indicator of a wrong prediction. Overall, across all time steps and all grid cells globally, the uncertainty is between $5.2 \times 10^{-6}\,\%$ and $37.5\,\%$ of the estimated hazard value, with an average of $13.6\,\%$.

## 4 Discussion

We evaluate PHELS against the set of LSE that was also used to derive optimal parameter values for Equations 5-6, which describe the signature relationships (Vrugt and Sadegh, 2013) between predictor variables and $\mathcal{H}$. This fit was hence based on a long-term and spatially-aggregated frequency distribution of LSE and not on individual LSE conditions, and has the advantage of robustness because a maximum amount of data was used. In the subsequent evaluation, the match between individual simulated and reported LSE and noLSE is measured in terms of e.g. AUC values, and this focuses on different aspects than the fit on a summary distribution of LSE. A similar procedure has been followed by Calvello and Pecoraro (2019) and Guzzetti et al. (2007) to obtain and evaluate their probabilistic respectively intensity-duration thresholds. Other options could have been to separate the LSE data into a training and a test subset as was done by Stanley et al. (2021), but this would reduce the number of data points both for fitting and validation.

PHELS performance increased when accounting for possible errors in the dating of LSE as well as time zone shifts (LSE3), i.e. PHELS often simulates hazard peaks within 3-day windows around LSE, regardless of the predictor variable(s). This is in line with findings of Kirschbaum and Stanley (2018). For LSE3, $\mathcal{H}$ performance based on rainfall approaches that of *ARI7*, because offsets between the timing of short-term rainfall and LSE are less penalized. The $\mathcal{H}$ performance based on *rzmc* is least impacted by the extended time window, because *rzmc* is typically slowly varying in time (seasonal). For this assessment, noLSE were only sampled within grid cells of reported LSE to emphasize the temporal aspect in the evaluation. When sampling noLSE globally randomly (noLSEglobal), we find very good $\mathcal{H}$ performance for all predictor variables, i.e. global spatio-temporal patterns of hazard are equally well captured, reinforcing the quality of the (*LSS*) estimates by Felsberg et al. (2022b).

As hydrological predictor variables we investigated daily rainfall, *ARI7* from MERRA-2 and *rzmc* from CLSM. Other datasets could be used (e.g. *rzmc* estimates from satellite-based data assimilation, as in Felsberg et al. (2021); Stanley et al. (2021)), or the predictors could be preprocessed differently, e.g. into daily rainfall maximum, antecedent soil moisture (Mirus et al., 2018), soil moisture changes (Wicki et al., 2020), or short- and long-term anomaly values. Percentile thresholds could moreover be derived for shorter time periods (seasonal, pentad), allowing for intra-seasonal hazard estimations. Note that if the connection of any of these predictor variables with LSE would differ from the exponential behaviour found for all predictor variables used in this study, the form of the fitted Equations 5-6 would need to be adapted. Furthermore, when multiple time-varying hydrological predictors are used, it could be recommended to implement the normalization factor in Equation 3: given the use of percentile values for the time-varying predictors, the probability for a single predictor variable is uniform (and normalization would not alter the results), but the joint probability of multiple percentile predictors is not uniform (and would alter the temporal behavior of PHELS).



PHELS can combine any hydrological variables or even other triggering sources (e.g. seismic). Inclusion of a third predictor variable changed $\mathcal{H}$ output characteristics from exponential to quasi-normal (see Appendix A), which allowed the use of a simple ensemble standard deviation as a metric for the $\mathcal{H}$ uncertainty. Any further or renewed extension requires a check of distribution characteristics and statistical measures should be chosen accordingly.

The $\mathcal{H}_{ens}$ uncertainty in terms of ensemble standard deviation follows a parabolic pattern and a linear upward trend with
increasing $\overline{\mathcal{H}}$. The first is induced by the characteristics of *rzmc* and *LSS*, which also show parabolic relationship between ensemble average and standard deviation (Felsberg et al., 2022b). The second reflects the behaviour of rainfall and its standard deviation. However, the uncertainty decreases again for very large $\overline{\mathcal{H}}$ which is likely a result of the design and boundedness of the input variables $x_h$ (percentiles). The sampling (and resampling) of the predictor variable values at the lower (upper) edge of the definition range, i.e. at percentile 1 (percentile 100), will inevitably lead to smaller ensemble standard deviations.

The spatial resolution in this study (36 km) is much coarser than a typical landslide extent. This helps to improve (not degrade) the hazard estimates, because it is easier to estimate the chance for a landslide in a large pixel than at a specific location (scaling effect). The 36-km simulated $\mathcal{H}$ thus describes spatio-temporal patterns of landslide probability for a larger area rather than for a single slope. Spatial variation of $\mathcal{H}$ within one grid cell extent can be expected where the environment is very heterogeneous and this will be partly captured in the $\mathcal{H}_{ens}$ uncertainty. The temporal evolution of $\mathcal{H}$, on the other hand,
is governed by hydrology and meso-scale meteorology, which are attributed with autocorrelation lengths of up to 40 km even in strongly mountainous terrain such as the Swiss Alps (Mittelbach and Seneviratne, 2012). A 36-km grid cell time series will thus adequately represent the temporal $\mathcal{H}$ variability.

The comparison of hydrological predictor variables shows that PHELS based on *ARI7* or rainfall performs best when rainfall is above average (above $50^{th}$ percentile, not shown), and PHELS based on *rzmc* performs best when *rzmc* is above average.
As a compromise, the hydro-meteorological approach using *rzmc*&rainfall performs slightly worse than *rzmc* (or rainfall) alone for elevated *rzmc* (rainfall), but much better for conditions of dry soil (no rainfall). This is also visible in the stratified AUC analysis, where AUC values range much closer together for *rzmc*&rainfall. Including *rzmc* on top of rainfall as predictor variable specifically improves the performance in spring when other water sources (meltwater, etc.) may be present. We found that including *rzmc* reduced missed alarms (on wet soils even small rainfall events may induce an LSE) and false alarms (where
the soil is dry, LSE occurrence is less likely) for the full data set. The reduction in false alarms was also reported by Ponziani et al. (2012); Segoni et al. (2018a); Stanley et al. (2021). While the number of missed alarms is lower for PHELS based on rainfall&*rzmc* than for PHELS based on rainfall alone, it is even lower for PHELS based on *ARI7*. A possible reason for this can be shallow landslides that are usually triggered by intensive rainfall irrespective of the soil moisture.

PHELS can be compared to different versions of LHASA (Kirschbaum and Stanley, 2018; Stanley et al., 2021), while keep-
ing in mind the large discrepancy in spatial resolution (36 km vs. 1 km), ratio of LSE to noLSE sampling (1:1 vs. roughly 1:10000), method of LSE and noLSE sampling (see LSE3, noLSEglobal) and the thresholds in probabilistic $\mathcal{H}$ where applicable. The following skill values for PHELS and LHASA refer to an evaluation of the hazard at the reported day of LSE. The first setup of LHASA combined *ARI7* and *LSS* into categorical 'nowcasts' (Kirschbaum and Stanley, 2018), and is comparable to PHELS using *ARI7* as hydrological predictor variable. The latter yields a TPR of 0.49, which is above the range of



reported TPR for LHASA (0.13 for an earlier version of GLC, 0.23 for LRC, and 0.38-0.45 with additional large-scale rain-
       fall event inventories) (Stanley et al., 2021). LHASA version 2.0 has a different setup (probabilistic combination of different
       input variables, among these slope, rainfall, soil moisture, see Stanley et al. (2021)), which is comparable to PHELS based on
       *rzmc*&rainfall. For the latter, we find a TPR of 0.46, whereas Stanley et al. (2021) find TPR values between 0.17 (GLC, also
       used as training data), 0.32 (LRC, test data) and up to 0.93 with additional inventories. For PHELS, using the $90^{th}$ temporal

percentile within a grid cell as a threshold results in 10 % positive predictions. Because landslides are rare events, the FPR also
       ranges close to this (0.1) but may differ for temporal subsets (see Table 2). This FPR is much higher than those for LHASA
       predictions (between 0.002 and 0.01, Kirschbaum and Stanley (2018); Stanley et al. (2021)). For both models, FPR might
       however be erroneously high due to known underreporting in the GLC.

## 5    Conclusions

In this study, we create the global Probabilistic Hydrological Estimation of LandSlides (PHELS) model, which produces daily
       landslide hazard [-] ($\mathcal{H}$) estimates at a coarse 36-km resolution. The PHELS model combines landslide susceptibility (*LSS*) and
       (percentiles of) hydrological predictor variables such as rainfall, a 7-day antecedent rainfall index [mm] (*ARI7*) or root-zone
       soil moisture content [$\mathrm{m}^3/\mathrm{m}^3$] (*rzmc*). Apart from deterministic $\mathcal{H}$ simulations, PHELS supports landslide hazard ensemble
       ($\mathcal{H}_{ens}$) simulations based on repeated sampling of *LSS* and the hydrological predictor variables. The resulting spread among

the ensemble members is a measure of the uncertainty of the simulations. To our knowledge, this is the first global landslide
       hazard model with uncertainty quantification. We conclude the following:

       1) deterministic $\mathcal{H}$ estimates with PHELS yield area under the ROC curve (AUC) values above 0.68, with the best perfor-
       mance (AUC=0.79) based on the combination of rainfall and *rzmc* as hydrological predictor variables, and second best based
       on *ARI7* (AUC=0.77). Including *rzmc* on top of rainfall can reduce missed alarms (especially during spring) and false alarms.

The performance of ensemble average $\mathcal{H}$ ($\overline{\mathcal{H}}$) is similar.

       2) $\mathcal{H}_{ens}$ uncertainty follows the behaviour of the input variable uncertainties (rainfall, *rzmc* and *LSS*), and is about 13.6 %
       of the daily simulated $\overline{\mathcal{H}}$ value globally and for the study period. The uncertainty follows a parabolic pattern introduced by the
       characteristics of *rzmc* and *LSS* where rainfall uncertainty is small and a positive linear relationship where rainfall uncertainty
       is large. Overall, the uncertainty of $\mathcal{H}$ simulations is small for very low and very high ensemble average $\mathcal{H}$, and larger for

intermediate values.

       The PHELS model is a flexible framework that allows the inclusion of other hydrological predictors or data sources (satellite
       data products, data assimilation) in future research. The approach can also be promising at smaller scales with local (in-
       situ) data, and offers a scaleable way to propagate uncertainties of various contributing predictors to traceable $\mathcal{H}$ uncertainty
       estimates.



*Code and data availability.* Code of the PHELS model setup can be found here: https://doi.org/10.5281/zenodo.7194280, hazard output in netcdf format for deterministic results (ARI7, rainfall, rzmc, rzmc&rainfall) and ensemble results (rzmc&rainfall) can be found here: https://doi.org/10.5281/zenodo.7188355. Code for the figures can be obtained upon request to the contact-author.

*Video supplement.* An animation of global ensemble average hazard (rzmc&rainfall) for the year 2015 can be found here: https://doi.org/10.5281/zenodo.7882809

*Author contributions.* AF designed the PHELS setup, created the code and conducted the analysis, supervised by GDL and JP along the way. GDL provided scientific guidance for all steps of this study, with special focus on the land surface model simulations and statistics. ZH provided expertise concerning the probabilistic statistics. JP and TS provided topical expertise for interpretation of results. All co-authors provided guidance on the study's content and contributed to the paper.

*Competing interests.* The authors declare that they have no conflict of interest.

*Financial support.* Anne Felsberg was funded by the Fonds Wetenschappelijk Onderzoek (grant no. FWO-G0C8918N). VSC usage was funded by KU Leuven (C14/16/045), FWO (1512817N) and the Flemish Government.

*Acknowledgements.* We thank Luca Brocca and Matthias Vanmaercke for being part of the PhD advisory committee of AF and encouraged the investigation of different predictor variables. The computational resources (high-performance computing) and services used in this work were provided by the VSC (Flemish Supercomputer Center).

## Appendix A: Distribution comparison

In addition to the ROC analysis, we evaluate the performance of deterministic PHELS for all hydrological predictor variables (*rzmc*, rainfall, *ARI7* and *rzmc*&rainfall) in terms of $\mathcal{H}$ distributions for LSE and noLSE. Specifically, we calculate the area between the quantile-quantile (QQ) line and the bisector ($A_{QQ}$) normalized by the total area underlying the bisector, as described in Felsberg et al. (2021). The larger $A_{QQ}$, the more different the two distributions. A large difference between the distributions implies that the variable in question is able to distinguish well between LSE and noLSE conditions, as discussed in Felsberg et al. (2021). To better highlight the long-term capability of PHELS, we also analyze $\mathcal{H}$ relative to the temporal average within a grid cell:



$$\mathcal{H}_{rel} = \frac{\mathcal{H} - \mathrm{avg}_t(\mathcal{H})}{\mathrm{max}_t(\mathcal{H}) - \mathrm{min}_t(\mathcal{H})}, \tag{A1}$$

where $\mathrm{avg}_t$, $\mathrm{min}_t$ and $\mathrm{max}_t$ denote the temporal average, minimum and maximum for this grid cell across the complete time period. For an assessment of the short-term capability, timesteps are limited to $\pm15$ days (around the LSE or noLSE in question) across all years to obtain an average, minimum and maximum $\mathcal{H}$ within this time window. This relative $\mathcal{H}$ is referred to as $\mathcal{H}_{rel,15}$.

**Table A1.** Differences between LSE and noLSE distributions of hazard values for *PHELS* based on different hydrological variables (rainfall, *ARI7*, *rzmc*, *rzmc*&rainfall) measured in terms of normalized $A_{qq}$. The corresponding distributions are shown in Figure A1. Results are shown for the deterministic $\mathcal{H}$, $\mathcal{H}$ relative to the temporal range within a grid cell ($\mathcal{H}_{rel}$), and relative to the long-term average hazard for +/-15 days ($\mathcal{H}_{rel,15}$).

| $x_h$ | **$A_{qq}$ based on** | | |
|---|---|---|---|
| | $\mathcal{H}$ | $\mathcal{H}_{rel}$ | $\mathcal{H}_{rel,15}$ |
| **rainfall** | 0.60 | 0.76 | 0.42 |
| **ARI7** | 0.65 | 0.83 | 0.48 |
| **rzmc** | 0.58 | 0.75 | 0.36 |
| **rzmc&rainfall** | 0.24 | 0.60 | 0.39 |

Extracting $\mathcal{H}$ for LSE and noLSE results in distributions as displayed in Figure A1. It illustrates that the inclusion of a second hydrological predictor variable changes results from exponential distributions to quasi-normal distributions. While differences between the distributions of LSE and noLSE can be captured by $A_{QQ}$, it is difficult to compare $A_{QQ}$ across such different types of distributions. Table A1 summarizes $A_{QQ}$ of all predictor variables for $\mathcal{H}$, $\mathcal{H}_{rel}$ and $\mathcal{H}_{rel,15}$. *ARI7* shows highest $A_{QQ}$, i.e. discrimination ability, for $\mathcal{H}$ predictions based on one hydrological predictor variable and *rzmc* has lowest $A_{QQ}$. $A_{QQ}$ is generally higher (lower) for $\mathcal{H}_{rel}$ ($\mathcal{H}_{rel,15}$) than for $\mathcal{H}$. For PHELS based on *rzmc*&rainfall, $A_{QQ}$ is smaller than for single variable PHELS. At the same time, $A_{QQ}$ increases for both $\mathcal{H}_{rel}$ and $\mathcal{H}_{rel,15}$, indicating improvements in the discrimination abilities both for long- and short-term patterns.



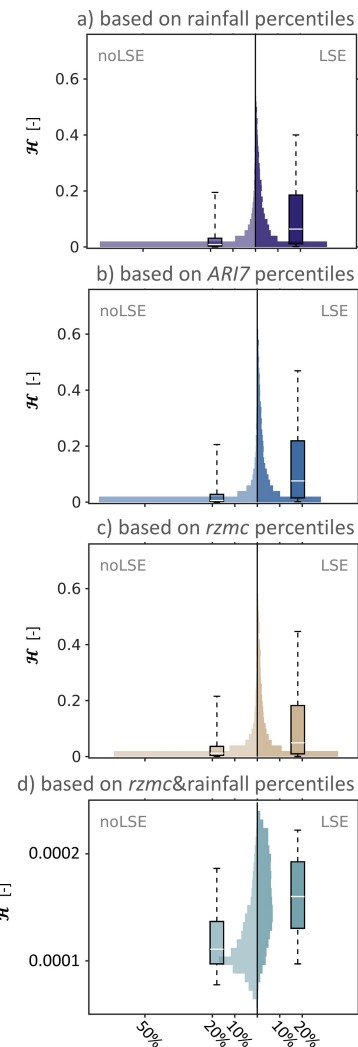

**Figure A1.** Distribution of PHELS $\mathcal{H}$ for known LSE (right, dark value) and sampled noLSE (left, light color), shown in histograms (background) and boxplots (foreground). Distributions are shown for PHELS based on a) rainfall percentiles, b) *ARI7* percentiles, c) *rzmc* percentiles ($x_h$ in Equation 5) and d) *rzmc*&rainfall percentiles ($x_{h1}$ and $x_{h2}$ in Equation 6).



**Abbreviations**

$A_{QQ}$  area between the quantile-quantile (QQ) line and the bisector.

$\mathcal{H}$  landslide hazard [-].

$\mathcal{H}_{ens}$  landslide hazard ensemble.

$\overline{\mathcal{H}}$  ensemble average $\mathcal{H}$.

*ARI7*  7-day antecedent rainfall index [mm].

*ARI*  antecedent rainfall index.

*LSS*  landslide susceptibility.

*rzmc*  root-zone soil moisture content [m$^3$/m$^3$].

**AUC**  area under the ROC curve.

**CLSM**  Catchment Land Surface Model.

**DJF**  December-January-February.

**EASEv2**  Equal-Area Scalable Earth version 2.

**FPR**  false positive rate.

**GLC**  Global Landslide Catalog.

**JJA**  June-July-August.

**LHASA**  Landslide Hazard Assessment for Situational Awareness.

**LRC**  Landslide Reporter Catalog.

**LSE**  landslide event.

**LSM**  land surface model.

**MAM**  March-April-May.

**MERRA-2**  Modern-Era Retrospective analysis for Research and Applications, Version 2.

**noLSE**  no landslide event.

**PHELS**  Probabilistic Hydrological Estimation of LandSlides.

**ROC**  Receiver Operation Characteristic.

**SON**  September-October-November.

**TPR**  true positive rate.



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
