# Peer review of "Probabilistic Hydrological Estimation of LandSlides (PHELS): global ensemble landslide hazard modelling"

_EGUsphere, 2023_

## Referee Comment (RC1)

This manuscript concisely presents a new approach to probabilistic landslide hazard modeling, which leverages a susceptibility model published previously in NHESS (Felsberg et al., 2022) with ensemble modeling to evaluate the suitability of a few alternative predictor variables for hydrologically triggered landslides. This PHELS model is applied at the global scale with coarse spatial and temporal resolution. Of the hydrologic predictors considered, rainfall with root zone soil moisture performed better than either of those variables alone and slightly better than an antecedent rainfall index used in the LHASA model (Stanley et al., *Frontiers*, 2021). When using a sparse global landslide catalogue, the output compares favorably to this existing model and the spatial and temporal variability in performance is shown, which reveals that uncertainty is lower during wetter seasons than drier ones. Furthermore, the probabilistic analysis reveals that very high and very low hazard predictions are well constrained, whereas the combinations of moderate susceptibility and moderate triggering conditions exhibit far greater uncertainties in landslide hazard predictions.

Overall, the topic is of considerable interest to NHESS readers, and this is a nice piece of work that presents several notable contributions, which ultimately warrants publication. Specifically, the approach for incorporating uncertainty in spatial and temporal probability of landsliding is novel and broadly applicable to hazard modeling, the evaluation of multiple predictor variables for hydrologic triggering is interesting, and the seasonal and spatial analysis as well as comparison of results to other global-scale analyses is useful. While these results are not particularly surprising and follow somewhat logically from the methods and input data, the work is technically sound and a useful reference. Methods and data are clearly described, such that results should be readily reproducible and the methods applied successfully with other data. The primary areas for improvement are largely editorial and include adding a more involved interpretation and context for the results, as well as some minor details in the presentation and figures. Therefore, the paper should be published after undergoing some minor revisions to address the following general and specific comments.

In general, the authors should make a little more effort to discuss what is interesting and useful about the results in the context of other studies. Readers should be confronted with both the advantages of the approach and its application, as well as its limitations, so that the utility and value of the resulting PHELS model is more apparent. For example, the framework for uncertainty assessment provides a robust approach to integrate forecast uncertainty that would be an important methodological advance for regional-scale landslide warning, but at the same time the global application seems of limited value for practical implementation locally and the reasonably strong performance is likely linked to the coarse spatial and temporal resolution and the decision to exclude a 6-day window for selecting non-triggering rainfall conditions. At the same time, the finding that the combination of daily rainfall with root zone soil moisture is more effective than seven-day rainfall index used in the LHASA model is consistent with recent advances in local landslide warning that leverage in-situ monitoring in favor of antecedent rainfall for reducing failed and false alarms, so it is useful to know that this potentially applies globally. Lastly, the revised version should include some discussion of why the model at such coarse resolution is useful, how that affects performance, and the limits of that utility.

In terms of presentation, the writing is clear and most of the figures are great. To address these and the general comments above, I have included the following specific comments and suggestions by line number:

L18. Specify: "… a two-step approach that separately evaluates where and when landsliding will occur." Also, why are you referencing the first approach (Stanley et al papers) and not a long list of others using the two-step, particularly since that is your approach here?

L23. It is misleading to imply that LHASA is specifically for landslide early warning since the rainfall data includes some latency and can at best be considered a "now-cast" of potential landslide conditions (i.e., see title of Stanley et al., 2021). At this point assessments that combine when and where landslides are likely cannot be used in a predictive mode for warning and their value for real-time hazard assessments is unclear. Furthermore, these are not typically ideal hazard assessments, in that they do not explicitly account for the magnitude and mobility of the potential landsliding, which is a significant consideration.

L25. Maybe it's worth mentioning that in addition to relying on a single threshold for landslide initiation (e.g., the 95%-ile of the ARI7 predictor at each grid cell), these also employ some susceptibility threshold that also amounts to a simple binary yes/no for landslide occurrence (though different thresholds of moderate or high have been used for different applications of the model). The value of your study is combining both within a probabilistic framework for both potential and initiation, so highlighting that here is worthwhile even if it's stated elsewhere.

L27. Whiteley et al (2019) is a very nice review of geophysical methods for landslide monitoring, but it's not exactly an appropriate reference for physically based modeling of landslide initiation potential. If I understand correctly, there are a lot of more appropriate examples (TRIGRS, Baum et al., *JGR-ES*, 2010; SCOOPS3D, Brien and Reid, *Rev. Eng. Geol*, 2008; SHALSTAB, Montgomery and Dietrich, *WRR*, 1994; etc., or review of this in the textbook Lu and Godt, 2013).

L45. Ok, here you define your limits of hazard modeling, but potentially should acknowledge that an ideal hazard assessment also includes the magnitude and mobility to determine where hazards are present, but in my view that's acceptable since at such coarse resolution that's not really as relevant.

L60. Here the use of "magnitude" is misleading. Be more specific and careful to clarify that you are evaluating the degree to which the spatiotemporal potential for landsliding is related to the uncertainty. There are lots of different ways you could consider "magnitude" of the event, from the size and velocity of the landslides to the number or extent of landslides or even their severity.

L71. Isn't there also a bias towards landslides in areas where they have an impact (i.e., roads, developed areas), as well as for landslide types that have a more notable impact (e.g., debris flows and shallow mass movements that affect infrastructure)?

L100. This seems to imply that most landslides (in the inventory?) fail at <1m depth. Is that supported by analysis of the GLC? Again, see comment earlier about dataset bias: is the GLC indeed mostly shallow landslides?

L191. This is a major assumption that warrants further discussion. Even if others have used this +/- 3days method before, I'm not convinced it is appropriate since you don't need a model to predict landslides when it's not raining hard. It seems to undermine the value of the model predictions by weighting the non-predictions to times when it's not really raining. A landslide hazard assessment tool really needs to be able to distinguish between triggering and non-triggering conditions *when it's actually raining hard* which your uncertainty assessments show it struggles most with in all but the most extremely high and low hazard levels.

Figure 4. The greater Seattle Area experienced widespread landsliding in mid-late January 2016, including several mentioned explicitly in our paper (Mirus et al., *Landslides*, 2018) and I think also elsewhere (see Luna and Korup, *GRL*, 2022). This is an interesting opportunity to show an example where the inventory is not just incomplete in space, but also in time and how that influences results.

Separately, this figure does a nice job of visualizing why a combination of RZSM and daily rainfall is a valuable combination.

L238. This is more interesting in that it shows that the cumulative rainfall variables are not necessarily the best predictors for landslide triggering, instead there are likely some sub-daily rainfall characteristics (e.g., three-hour rainfall intensity, see Patton et al., *NHESS,* 2023). In contrast, the predictors that integrate the hydrologic variable, RZSM, may better capture those, albeit not explicitly.
Still, relying on a +/- 3-day window to obtain great model performance really undermines some of the potential utility of such a tool. Emergency planners and the general public don't really want to be on high alert ready to take action for 6 days.

L265. Can the discussion include any conjecture on why these spatial variability in uncertainty? Is it all due to the triggering not capturing the type of landsides, is it greater combined uncertainty in both the susceptibility and initiation? Is it data limitations?

Figure 10b. This figure is visually very challenging and not particularly accessible for individuals with red-green color blindness. Can you further reinforce this with a better color scheme and show TN, FP, FN, and FP with distribution curves of H and standard deviation on the x and y axes, respectively?

[Figure]

L285. Is this accounting for errors the +/- 3 day window? Again, this undermines the practical value of this type of model if it can only perform well at capturing a landslide event within a 6-day window, particularly given the coarse spatial resolution.

L330. Yes, we also found replacing antecedent rainfall with antecedent soil moisture decreased failed and false alarms (Mirus et al., *Landslides*, 2018).

L331. I don't quite follow this logic. ARI7 uses a weighted averaging daily rainfall over 7 days... how does that capture sub-daily bursts in rainfall intensity that may trigger landslides?

L334-345. Interesting, but I would think that with the same dataset a coarser resolution model would likely perform better, particularly for FPR and particularly for isolated landslides.

L346-348. Yes, the GLC is incomplete both spatially and temporally, see previous comment about Seattle-area landslides in January 2016. Are there areas where it is more or less complete that you could compare to assess this?

L349. The discussion section would benefit from presenting the sources of uncertainty are not considered and which of those potentially could be included. You are able to consider the uncertainty in susceptibility and triggering conditions since they can be quantified. However, the uncertainty due to incomplete inventories in space and time is not considered and could only be done if there were appropriate data to support this. Conversely, could the framework integrate weather forecasts and incorporate uncertainty in those forecasts relative to triggering conditions identified with the MERRA2 data?

P.S. The animation is a nice bonus of this paper.

---

## Author Comment (AC1)

The authors thank the reviewers for their constructive comments. The comments are shown in regular fonts (we added numbers), ***our responses are in bold italic, blue fonts***. *Changes made in the manuscript are printed in italic, underlined, blue fonts.* ***Our line references refer to the updated manuscript with track-changes.***

Reviewer #1: Mirus, Ben

This manuscript concisely presents a new approach to probabilistic landslide hazard modeling, which leverages a susceptibility model published previously in NHESS (Felsberg et al., 2022) with ensemble modeling to evaluate the suitability of a few alternative predictor variables for hydrologically triggered landslides. This PHELS model is applied at the global scale with coarse spatial and temporal resolution. Of the hydrologic predictors considered, rainfall with root zone soil moisture performed better than either of those variables alone and slightly better than an antecedent rainfall index used in the LHASA model (Stanley et al., *Frontiers*, 2021). When using a sparse global landslide catalogue, the output compares favorably to this existing model and the spatial and temporal variability in performance is shown, which reveals that uncertainty is lower during wetter seasons than drier ones. Furthermore, the probabilistic analysis reveals that very high and very low hazard predictions are well constrained, whereas the combinations of moderate susceptibility and moderate triggering conditions exhibit far greater uncertainties in landslide hazard predictions.

Overall, the topic is of considerable interest to NHESS readers, and this is a nice piece of work that presents several notable contributions, which ultimately warrants publication. Specifically, the approach for incorporating uncertainty in spatial and temporal probability of landsliding is novel and broadly applicable to hazard modeling, the evaluation of multiple predictor variables for hydrologic triggering is interesting, and the seasonal and spatial analysis as well as comparison of results to other global-scale analyses is useful. While these results are not particularly surprising and follow somewhat logically from the methods and input data, the work is technically sound and a useful reference. Methods and data are clearly described, such that results should be readily reproducible and the methods applied successfully with other data. The primary areas for improvement are largely editorial and include adding a more involved interpretation and context for the results, as well as some minor details in the presentation and figures. Therefore, the paper should be published after undergoing some minor revisions to address the following general and specific comments.

***We thank the reviewer for the positive feedback and encouraging words.***

1) In general, the authors should make a little more effort to discuss what is interesting and useful about the results in the context of other studies. Readers should be confronted with both the advantages of the approach and its application, as well as its limitations, so that the utility and value of the resulting PHELS model is more apparent. For example, the framework for

uncertainty assessment provides a robust approach to integrate forecast uncertainty that would be an important methodological advance for regional-scale landslide warning, but at the same time the global application seems of limited value for practical implementation locally and the reasonably strong performance is likely linked to the coarse spatial and temporal resolution and the decision to exclude a 6-day window for selecting non-triggering rainfall conditions. At the same time, the finding that the combination of daily rainfall with root zone soil moisture is more effective than seven-day rainfall index used in the LHASA model is consistent with recent advances in local landslide warning that leverage in-situ monitoring in favor of antecedent rainfall for reducing failed and false alarms, so it is useful to know that this potentially applies globally. Lastly, the revised version should include some discussion of why the model at such coarse resolution is useful, how that affects performance, and the limits of that utility.

*We extended the Discussion to provide more insights into advantages, limitations and applicability of the PHELS model and its setup, which was also suggested by reviewer #2, Clàudia Abancó. The new paragraphs are therefore a combination of recommendations:*

*This known incompleteness of the inventory not only influences the performance evaluation, but also adds to the uncertainty of the model fitting process (i.e. Equations 5-6). While the goodness of fit can be quantified (see Table 1) and theoretically propagated, it is still relative to the available inventory. Quantification of inventory-induced uncertainty requires very detailed or synthetic landslide inventories and has been subject of many studies for LSS (Steger et al., 2017; Lin et al., 2021) but less so for hazard assessment. PHELS does not account for such inventory-induced uncertainty, but it does include the uncertainties and within-grid-cell heterogeneity of input variables by using an ensemble approach. The latter allows to easily account for, e.g., the uncertainty in rainfall, which is directly available from ensemble weather prediction systems. Or to account for modeled soil moisture uncertainties, which can be obtained from ensemble land surface model simulations that are usually optimized to match the variations in observations. Nevertheless, models are always a simplification of real world conditions and the downscaling of coarse-scale model estimates to fine scale applications remains a challenge.*

*PHELS provides reliable insights into spatio-temporal patterns of landslide hazard but has limitations in the context of actual early warning systems. These usually require higher spatial resolution and temporal accuracy. The coarse spatial resolution would hence call for downscaling methods to obtain within grid-box distributions. And although we use the evaluation approach LSE3 because of time shifts and possible observation errors, the fact that peaks of hazard are often simulated within a 3-day window around a recorded LSE may also indicate a low temporal accuracy, which might be mainly associated with the coarse-scale global re-analysis input of precipitation.. For early warning systems the question moreover remains how to interpret or use the hazard uncertainty. Low enough uncertainty could be used as a secondary condition before warnings are issued to the public or the uncertainty could be directly communicated as is. However, ensemble measures such as the maximum predicted hazard ("worst case scenario") or the 90[th] quantile of ensemble hazard prediction might be easier to understand. While this study used PHELS with specific spatio-temporal resolution and input data, its adjustable, modular character makes PHELS a general framework for hazard*

*estimation that can be tailored to specific purposes. If adequate landslide and hydrological data are available, it would therefore be possible to create a PHELS setup that is more suitable for local to regional landslide early warning systems. (Lines 375-397)*

In terms of presentation, the writing is clear and most of the figures are great. To address these and the general comments above, I have included the following specific comments and suggestions by line number:

2) L18. Specify: "… a two-step approach that separately evaluates where and when landsliding will occur." Also, why are you referencing the first approach (Stanley et al papers) and not a long list of others using the two-step, particularly since that is your approach here?

*We included your suggested extension in the manuscript. We had not included references for the two-step process here, since the following paragraphs elaborate this two-step process in more detail and provide references. However, to avoid confusion about this, we included the references in question already here. […] or in a two-step process that separately evaluates where and when landsliding is likely to occur (Kirschbaum and Stanley, 2018; Monsieurs et al., 2019a, b; Bordoni et al., 2020). (Lines 18-19)*

3) L23. It is misleading to imply that LHASA is specifically for landslide early warning since the rainfall data includes some latency and can at best be considered a "now-cast" of potential landslide conditions (i.e., see title of Stanley et al., 2021). At this point assessments that combine when and where landslides are likely cannot be used in a predictive mode for warning and their value for real-time hazard assessments is unclear. Furthermore, these are not typically ideal hazard assessments, in that they do not explicitly account for the magnitude and mobility of the potential landsliding, which is a significant consideration.

*This is a valid point. We added a separate reference for the use of susceptibility maps in early warning systems and give now-casting as another use-case to introduce the mention of LHASA in the next sentence. Others are specifically developed to be used in a landslide early warning system (Guzzetti et al., 2020) or 'now-casting' approach: [...] (Lines 24-25)*

4) L25. Maybe it's worth mentioning that in addition to relying on a single threshold for landslide initiation (e.g., the 95%-ile of the ARI7 predictor at each grid cell), these also employ some susceptibility threshold that also amounts to a simple binary yes/no for landslide occurrence (though different thresholds of moderate or high have been used for different applications of the model). The value of your study is combining both within a probabilistic framework for both potential and initiation, so highlighting that here is worthwhile even if it's stated elsewhere.

*This is a good point. We added the keywords "categorized" and "thresholds" to the sentence to underline this difference. [...]: the global, categorized LSS assessment by Stanley and Kirschbaum (2017), for instance, has been developed to allow severity thresholds within the first version of the Landslide Hazard Assessment for Situational Awareness (LHASA) model (Kirschbaum and Stanley, 2018). (Lines 25-28)*

5) L27. Whiteley et al (2019) is a very nice review of geophysical methods for landslide monitoring, but it's not exactly an appropriate reference for physically based modeling of landslide initiation potential. If I understand correctly, there are a lot of more appropriate examples (TRIGRS, Baum et al., *JGR-ES*, 2010; SCOOPS3D, Brien and Reid, *Rev. Eng. Geol*, 2008; SHALSTAB, Montgomery and Dietrich, *WRR*, 1994; etc., or review of this in the textbook Lu and Godt, 2013).

*Thank you for catching this inconsistency! We added the review by Lu and Godt, as well as the example of TRIGRS and removed the reference to Whiteley et al.: The temporal probability can either be calculated explicitly by physical models that compute the shear strength and stress in slopes (Lu and Godt, 2013; Baum et al., 2010) [...] (Lines 29-30)*

6) L45. Ok, here you define your limits of hazard modeling, but potentially should acknowledge that an ideal hazard assessment also includes the magnitude and mobility to determine where hazards are present, but in my view that's acceptable since at such coarse resolution that's not really as relevant.

*Indeed, the question of what hazard modelling should comprise is scale dependent. We added a subclause referring to this: We comprise all of the above-mentioned approaches under the term 'hazard modelling' , while being aware that at smaller scales the size and mobility of a landslide may also be an essential part of hazard prediction. (Lines 49-51)*

7) L60. Here the use of "magnitude" is misleading. Be more specific and careful to clarify that you are evaluating the degree to which the spatiotemporal potential for landsliding is related to the uncertainty. There are lots of different ways you could consider "magnitude" of the event, from the size and velocity of the landslides to the number or extent of landslides or even their severity.

*Here, we simply intended to refer to the magnitude of the hazard value, not the magnitude of the landslide event. Thanks for pointing out that the phrasing can be misleading. We added "value" to avoid any misunderstanding: 2) Is the estimated uncertainty related to the magnitude of the simulated hazard value? (Line 66)*

8) L71. Isn't there also a bias towards landslides in areas where they have an impact (i.e., roads, developed areas), as well as for landslide types that have a more notable impact (e.g., debris flows and shallow mass movements that affect infrastructure)?

*True. We extended the sentence to reflect on this bias from the data collection method as well: Note that the known economic and English-language bias, as well as the fact that media reports tend to focus on inhabited areas and landslides with notable impact on infrastructure, will affect the completeness of these inventories and reduce the reliability of their 'absence reporting'. (Lines 75-78)*

*We also edited the caption of Figure 1 to make clear that these are only the reported landslides.*

9) L100. This seems to imply that most landslides (in the inventory?) fail at <1m depth. Is that supported by analysis of the GLC? Again, see comment earlier about dataset bias: is the GLC indeed mostly shallow landslides?

*The GLC uses a system of categorical size descriptors (small, medium, large, very large, catastrophic) depending on the estimated volume, i.e. it is difficult to retrieve information on the depth of the shear plane itself. Most landslides from the GLC are reported to be of "medium" size (>10m3, <1000m3). Essentially, what we intend to say is that rzmc is more suitable than the surface water content or the total water storage by themselves because 1m depth should be closer to the shear plane. We will rephrase the sentence as follows:*
*The rzmc contains information on both surface water content and groundwater and should therefore be indicative not only of water content at landslide shear planes <1m, which we consider a typical depth, but also for more shallow or deep-seated landslides. (Lines 106-109)*

10) L191. This is a major assumption that warrants further discussion. Even if others have used this +/- 3days method before, I'm not convinced it is appropriate since you don't need a model to predict landslides when it's not raining hard. It seems to undermine the value of the model predictions by weighting the non-predictions to times when it's not really raining. A landslide hazard assessment tool really needs to be able to distinguish between triggering and non-triggering conditions *when it's actually raining hard* which your uncertainty assessments show it struggles most with in all but the most extremely high and low hazard levels.

*This is a very valid point and actually already describes the reason why rzmc by itself is not the best of our tested predictor variables. For local to regional hazard models with very reliable landslide data and meteorological information, we agree, focus should lie on the exactness of the hazard prediction within a multi-day window. At the global scale and coarse spatial resolution, however, we are dealing with uncertainties in a) rainfall data, b) landslide timing from media reports, c) time zone shifts, i.e. PHELS is destined to capture general spatio-temporal patterns rather than concrete landslide events. We use reported landslide events as indicators for such patterns. Nevertheless, we included a sentence in the discussion reflecting on the limitations of the +/- 3days method (see second paragraph in reply to comment 1)*

11) Figure 4. The greater Seattle Area experienced widespread landsliding in mid-late January 2016, including several mentioned explicitly in our paper (Mirus et al., *Landslides*, 2018) and I think also elsewhere (see Luna and Korup, *GRL*, 2022). This is an interesting opportunity to show an example where the inventory is not just incomplete in space, but also in time and how that influences results.

Separately, this figure does a nice job of visualizing why a combination of RZSM and daily rainfall is a valuable combination.

*Thank you for reminding us of the dates of the landslides in your publication from 2018. We included this example in the discussion: For both models, FPR might however be erroneously high due to known underreporting in the GLC , even within well reported areas. Figure 4 for example misses mid-late January events of 2016 in the Seattle area that were reported by Mirus et al. (2018). (Lines 372-374)*

*Thanks also for pointing us to the publication by Luna and Korup, it is an interesting read and concept! We include the concept of seasonal or monthly landslide modelling in the discussion and conclusion as well: Other datasets could be used […], the predictors could be preprocessed differently, e.g. into daily or 3-hourly rainfall maximum (Patton et al.,*

*2023), monthly rainfall (Luna and Korup, 2022), antecedent soil moisture (Mirus et al., 2018), soil moisture changes (Wicki et al., 2020), or short- and long-term anomaly values. (Lines 305-309)*

*The approach can also be promising at smaller scales with local (in-situ) data or for seasonal modelling […]. (Lines 416-417)*

12) L238. This is more interesting in that it shows that the cumulative rainfall variables are not necessarily the best predictors for landslide triggering, instead there are likely some sub-daily rainfall characteristics (e.g., three-hour rainfall intensity, see Patton et al., *NHESS,* 2023). In contrast, the predictors that integrate the hydrologic variable, RZSM, may better capture those, albeit not explicitly. Still, relying on a +/- 3-day window to obtain great model performance really undermines some of the potential utility of such a tool. Emergency planners and the general public don't really want to be on high alert ready to take action for 6 days.

*It is true that PHELS was not developed with the needs of emergency planners in mind, and that our performance analysis lacks this viewpoint. In contrast to in-situ observations of rainfall close to the landslide location in question, precipitation data from a reanalysis model does come with larger possible error, as does our global approach. This was the reasoning behind the +/-3 days. We now discuss the limitation for early warning (see reply to comment 1 and 10)*

*Thanks also for pointing us to Annette Patton's recent preprint. We included this in our discussion of possible other predictor variables (see reply to comment 11)*

13) L265. Can the discussion include any conjecture on why these spatial variability in uncertainty? Is it all due to the triggering not capturing the type of landsides, is it greater combined uncertainty in both the susceptibility and initiation? Is it data limitations?

*From a pure modeling point of view, it is the uncertainty of the input variables and their combination. Situations where soil moisture and rainfall and LSS are high can easily be distinguished as highly hazardous, and hazard ensemble members deviate less from one another. The same goes for the opposite conditions as well. In addition, the ensembles of the input parameters (soil moisture, LSS) are generally smaller for more extreme magnitudes due to their boundedness. We added a paragraph discussing the concrete example of 15 September 2015 and also connect the observations to insights from Figure 10:* *This connection of $H_{ens}$ uncertainty with the uncertainty of the input variables generates a spatial pattern that is closely following the input patterns. As examples for low $H_{ens}$ uncertainty on 15 September 2015 we found central USA, the Amazon and the Congo basin (see Figure 8). These regions have low LSS and they exhibit dry conditions at this time (low rzmc and low or no rainfall) with small connected uncertainty: consequently, low $\bar{H}$ values and low $H_{ens}$ uncertainties are found. Since these regions have nearly no observed LSE (see Figure 1), sampled grid cells would probably be true negatives (TN). For the complete study period TN also showed lowest $\bar{H}$ and $H_{ens}$ uncertainty (see Figure 10b). As examples for high $H_{ens}$ uncertainty on 15 September 2015 we found Central America and China. Both record a large number of observed LSE and high LSS with low uncertainty. However, rzmc is intermediately high with increased uncertainty and rainfall is high with large uncertainty, resulting in $H_{ens}$*

*uncertainty also being high. Sampled grid cells would probably be positive predictions (TP, FP), which for the complete study period also showed highest $\bar{\bar{H}}$ and $H_{ens}$ uncertainty (see Figure 10b). (Lines 328-337)*

14) Figure 10b. This figure is visually very challenging and not particularly accessible for individuals with red-green color blindness. Can you further reinforce this with a better color scheme and show TN, FP, FN, and FP with distribution curves of H and standard deviation on the x and y axes, respectively?

*Thank you for this suggestion that indeed makes the plot more understandable! We changed the color scheme and added distribution curves of the groups along the x and y axes.*

[Figure]

*The caption of the Figure was updated accordingly: [...] The marginal distributions of the ensemble average and standard deviation are shown on the top and side panels. Median values are indicated by the larger symbols on top of the scatter cloud and on the marginal distributions. [...]*

*And we describe the different distributions in the Results section: Symbols and color indicate whether LSE and noLSE were correctly captured (true positive - TP; true negative TN) or not (false positive - FP; false negative - FN), again using the temporal 90th percentile of $\bar{\bar{H}}$ as a threshold. The distributions of $\bar{\bar{H}}$ for the positives are significantly different from that of the TN and they have a peaked distribution. Whereas the distribution for the FN is wider and largely overlaps with the distribution for the positives and the TN. While median uncertainty for FN, i.e. missed alarms is larger than that for TN, median uncertainty for FP, i.e. false alarms, is nearly identical to that of TP. (Lines 278-283)*

15) L285. Is this accounting for errors the +/- 3 day window? Again, this undermines the practical value of this type of model if it can only perform well at capturing a landslide event within a 6-day window, particularly given the coarse spatial resolution.

*We added a paragraph in the Discussion on the topic of the +/-3 day window (see reply to comment 10)*

16) L330. Yes, we also found replacing antecedent rainfall with antecedent soil moisture decreased failed and false alarms (Mirus et al., *Landslides*, 2018).

*Thanks for reminding us that you also found this in your study. We added the reference: The reduction in false alarms was also reported by Ponziani et al. (2012); Mirus et al. (2018); Segoni et al. (2018a); Stanley et al. (2021) . (Lines 353-354)*

*And also in the introduction: [...] the inclusion of soil water content has been found to prevent false alarms, independent of the data source (Ponziani et al. 2012; Mirus et al. 2018; Segoni et al. 2018a; Stanley et al. 2021). (Lines 43-45)*

17) L331. I don't quite follow this logic. ARI7 uses a weighted averaging daily rainfall over 7 days… how does that capture sub-daily bursts in rainfall intensity that may trigger landslides?

*We understand the confusion. What we intended to say is that intensive rainfall in the previous days (high ARI) may not have propagated deep enough into the soil (high rzmc) for a positive prediction resp. alarm based on rainfall&rzmc, and result in a missed alarm. We decided to shift the focus in the phrasing to the meaning of alarms based on ARI7, in accordance with a comment by reviewer #2 Clàudia Abancó: While the number of missed alarms is lower for PHELS based on rainfall&rzmc than for PHELS based on rainfall alone, it is even lower for PHELS based on ARI7. A possible reason for this can be that (intensive) antecedent rainfall prepares failure by progressively destabilizing the slope. (Lines 354-357)*

18) L334-345. Interesting, but I would think that with the same dataset a coarser resolution model would likely perform better, particularly for FPR and particularly for isolated landslides.

*FPR is defined as the ratio of false positives over all observed negatives, i.e. all noLSE in the reference data. Typically, noLSE are easier to predict because of e.g. absence of rain or very dry conditions and the amount of false positive predictions does not increase proportionally to the amount of sampled noLSE references. When moving from an LSE-noLSE sampling rate of 1:1 (PHELS) to 1:10000 (LHASA) the FPR is therefore by design much lower. We added this in the sentence: Due to the choice in threshold and the larger LSE-noLSE sampling ratio (see above) this FPR is by design much higher than those for LHASA predictions […] (Lines 370-371)*

19) L346-348. Yes, the GLC is incomplete both spatially and temporally, see previous comment about Seattle-area landslides in January 2016. Are there areas where it is more or less complete that you could compare to assess this?

*This is a good question. We had expected the GLC to be rather complete in the US, as compared to other regions worldwide, but your example of the Seattle-area illustrates that this is not the case. For spatial and temporal completeness, one would probably have to turn to a local or regional inventory. The question remains, however, how*

*complete landslide inventories can be, in general. The discussion now includes a paragraph on landslide-inventory-induced uncertainty. (see first paragraph in reply to comment 1)*

20) L349. The discussion section would benefit from presenting the sources of uncertainty are not considered and which of those potentially could be included. You are able to consider the uncertainty in susceptibility and triggering conditions since they can be quantified. However, the uncertainty due to incomplete inventories in space and time is not considered and could only be done if there were appropriate data to support this. Conversely, could the framework integrate weather forecasts and incorporate uncertainty in those forecasts relative to triggering conditions identified with the MERRA2 data?

*We included a paragraph in the discussion on the effect of inventory-induced uncertainty on the one hand, and uncertainty that PHELS takes into account on the other hand (see first paragraph in reply to comment 1).*

P.S. The animation is a nice bonus of this paper.

*Thanks!*

---

## Author Comment (AC2)

The authors thank the reviewers for their constructive comments. The comments are shown in regular fonts (we added numbers), *our responses are in bold italic, blue fonts*. *Changes made in the manuscript are printed in italic, underlined, blue fonts.* *Our line references refer to the updated manuscript with track-changes.*

Reviewer #2: Abancó, Clàudia

1) General Comments:

The main topic discussed in this manuscript is the uncertainty on the hazard estimation of hydrologically-triggered landslides. It presents a new model at global scale (PHELS), that estimates the daily hazard of hydrologically-triggered landslides at a coarse resolution (36 km) at the same time that it estimates its uncertainty by generating ensemble simulations. The paper is focused on the analysis of the performance of the temporal component (hydrological predictors) of the hazard estimation, as the static part (landslide susceptibility) is based in a paper already published (Felsberg et al., 2022).

The manuscript analyses the potential of three main hydrological predictor variables: the daily rainfall, the 7-day antecedent rainfall index (ARI7) and the root-zone soil moisture content (rzmc), although it does not go into detail on the uncertainty on the obtention of these values. Specially the rzmc is a factor that is very sensitive to the input parametres of the Catchment Land Surface Model (CLSM), as for exemple the soil porosity. Although my expertise is not in data-driven models, I assume that the results could also be affected by this sensitivity, therefore I'd recomment that authors acknowledge that some uncertainty could be induced by the source of the hydrological predictors.

The topic is interesting and novel, since as the authors point out, the literature on quantification of hazard uncertainty is very scarce, and only some attempts to quantify uncertainty of susceptibility or rainfall thresholds uncertainty have been published. In general, I think the authors should further emphasize the main advantages and limitations of PHELS compared to other models, such as the ones that don't provide uncertainty.

The paper is very well written, clear and, even if in some parts some clarification may help (at least for the non-experts in the topic), it is in general easy to read.

The conclusions are consistent with the evidence and arguments presented. They address the main questions proposed.

The Figures and Tables are in general clear, and helpful to follow the paper.

*We thank the reviewer for this positive feedback and the constructive comments. We clarified the manuscript following your specific comments, and extended the discussion to include a paragraph on uncertainty sources that are and are not taken into account, as well as to the applicability, advantages and limitations of the PHELS model framework. Since both were also suggested by reviewer #1, Ben Mirus, the new paragraphs are a combination of suggested discussion points:*

*This known incompleteness of the inventory not only influences the performance evaluation, but also adds to the uncertainty of the model fitting process (i.e. Equations 5-6). While the goodness of fit can be quantified (see Table 1) and theoretically propagated, it is still relative to the available inventory. Quantification of inventory-induced uncertainty requires very detailed or synthetic landslide inventories and has been subject of many studies for LSS (Steger et al., 2017; Lin et al., 2021) but less so for hazard assessment. PHELS does not account for such inventory-induced uncertainty, but it does include the uncertainties and within-grid-cell heterogeneity of input variables by using an ensemble approach. The latter allows to easily account for, e.g., the uncertainty in rainfall, which is directly available from ensemble weather prediction systems. Or to account for modeled soil moisture uncertainties, which can be obtained from ensemble land surface model simulations that are usually optimized to match the variations in observations. Nevertheless, models are always a simplification of real world conditions and the downscaling of coarse-scale model estimates to fine scale applications remains a challenge.*

*PHELS provides reliable insights into spatio-temporal patterns of landslide hazard but has limitations in the context of actual early warning systems. These usually require higher spatial resolution and temporal accuracy. The coarse spatial resolution would hence call for downscaling methods to obtain within grid-box distributions. And although we use the evaluation approach LSE3 because of time shifts and possible observation errors, the fact that peaks of hazard are often simulated within a 3-day window around a recorded LSE may also indicate a low temporal accuracy. For early warning systems the question moreover remains how to interpret or use the hazard uncertainty. Low enough uncertainty could be used as a secondary condition before warnings are issued to the public or the uncertainty could be directly communicated as is. However, ensemble measures such as the maximum predicted hazard ("worst case scenario") or the 90th quantile of ensemble hazard prediction might be easier to understand. While this study used PHELS with specific spatio-temporal resolution and input data, its adjustable, modular character makes PHELS a general framework for hazard estimation that can be tailored to specific purposes. If adequate landslide and hydrological data are available, it would therefore be possible to create a PHELS setup that is more suitable for local to regional landslide early warning systems. (Lines 376-398)*

Specific comments:

2) L29: binary approach: for the landslide hazard assessment or for the empirical temporal probability?

*Here, the binary approach refers to the temporal probability, thanks for pointing out that this was not clear. We merged the two paragraphs to emphasize that the second one still refers to the temporal probability. (Line 32)*

3) L34-40: do all these refer to the root zone or some include also lower layers of the soil?

*We use "soil moisture" in a general sense. The references use soil moisture at different levels and also from different data sources. Soil moisture information from satellites informs about the surface soil moisture (upper 5 cm of soil), in-situ observations reach*

*depths up to 1.40m, and studies that applied modelling use layers from surface to groundwater. We added this information in the manuscript. The measures of soil moisture range from antecedent soil moisture (Mirus et al., 2018; Wicki et al., 2020) and increase in soil saturation (Wicki et al., 2020) to soil moisture of the day (Bordoni et al., 2020), and refer to different soil layers (surface: Ponziani et al. (2012); Brocca et al. (2016); Thomas et al. (2019); Bordoni et al. (2020), root-zone: Brocca et al. (2016); Mirus et al. (2018); Thomas et al. (2019); Wicki et al. (2020), groundwater: Uwihirwe et al. (2022)). (Lines 39-43)*

4) L99: CLSM- what is the source of the inputs of the model (e.g.: soil porosity)? Also, in what units is rzmc?

*The input for CLSM-parameters of e.g. soil porosity comes from the U.S. General Soil Map (STATSGO2) and the Harmonized World Soil Database version 1.21, further details can be found in De Lannoy et al. 2014, and an overview is provided in Table 1 in Felsberg et al. 2022. The unit of rzmc is [m3/m3] as introduced in the introduction. To avoid future confusion, we also added the unit here in the methodology section. […] to simulate rzmc [ m3/m3 ] (0-100 cm) […] (Line 106)*

5) L125: as the temporally dynamic soil moisture...or also ARI7?

*Yes, indeed. We added this to the sentence. While LSS could conceptually be considered a prior probability we opt to use it as a temporally static (but spatially varying) variable and implement it as a condition in a similar way as the temporally dynamic soil moisture, ARI7 and rainfall. (Lines 131-133)*

6) Table 1: I am not sure if ARI7&rzmc were not tested because conceptually they both represent the same parameter (soil moisture)? If this is the case, I am not sure this is correct, as ARI7 may imply infiltration of water to lower levels (and consequent instabilization of the slope due to the water table rise) while rmzc is only for the upper layer of the soil.

*Indeed, this was our reasoning. Antecedent rainfall has in the literature often been used as a proxy for soil moisture and we expected the information content to be redundant. For land surface models it is known that lower level soil moisture and rzmc strongly correlate and the inclusion of total water storage in addition to rzmc could hence be expected to have negligible amount of new information. Since ARI however includes additional information on rainfall, it might be worthwhile to investigate the combination of ARI and rzmc. We added a sentence on this in the discussion: Furthermore different combinations of the predictors could be tested, e.g. rzmc and ARI7. (Lines 309-310)*

7) L186: As I understand, PHELS is trained with all the LSE for 2007-2020, and tested with the same data? Do you think this could induce some misleading results in the evaluation?

*Yes, we evaluate against the set of LSE that was also used to derive the parameter values for the PHELS equations. In the discussion we already have a paragraph explaining why we do not expect misleading results from this approach, and also give an outlook about possible other approaches (Line 287).*

8) Figure 4: What about the high H values in 2016? All the predictors show high H around 2016-02 but no LSE is recorded. Would they be false alarms or a limitation of the GLC?

*Theoretically, both are possible. Thanks to reviewer #1, Ben Mirus (see his comment 11), we however got reminded that the "Seattle Area experienced widespread landsliding in mid-late January 2016" (Mirus et al. 2018), i.e. it is a limitation of the GLC. We now mention this fact in the discussion: Figure 4 for example misses mid-late January events of 2016 in the Seattle area that were reported by Mirus et al. (2018). (Lines 373-374)*

9) Figure 6: This is an interesting Figure!

*Thanks for this feedback!*

10) Figure 8a: Again, going back to the rzmc absolute values. I can see values around 0.5 m3/m3 in SE Asia, that would correspond to soil porosity of 0.5 (considering that the soil is fully saturated). These are very high values for porosity, only typical for some sort of coarse sand or silt, but not common. I have noticed this in SMAP-L4 products derived from CLSM, and in my opinion is a limitation that the use of global models have. I would only make a point here to say that absolute values of rzmc may be overestimated due to this

*Thank you for sharing this insight. Indeed, global models lack accuracy, especially when it comes to soil characteristics and geology. With our approach to transform the absolute values of rzmc into percentiles according to the long-term climatology, we hopefully reduce the effect that a positive bias, i.e. a consistent overestimation, would have on landslide hazard estimation.*

11) L332: I think this could be because ARI7 is actually giving a larger picture of the mechanical process going on in the slopes and closely related to the instability process. So, I agree that soil moisture of the upper layer is not always a good indicator.

*This is a good point and we rephrased the sentence as follows: While the number of missed alarms is lower for PHELS based on rainfall&rzmc than for PHELS based on rainfall alone, it is even lower for PHELS based on ARI7 . A possible reason for this can be that (intensive) antecedent rainfall prepares failure by progressively destabilizing the slope. (Lines 354-357)*

12) Conclusions: as I said earlier, in the discussion/conclusions I miss some emphases on the advantages and limitations of PHELS over other models, i.e: the applicability of such a model

*We now included a paragraph on the advantages, limitations and applicability of PHELS in the discussion (see second paragraph in reply to comment 1)*